# First Study on the Oxidative Stability and Elemental Analysis of Babassu (*Attalea speciosa*) Edible Oil Produced in Brazil Using a Domestic Extraction Machine

**DOI:** 10.3390/molecules24234235

**Published:** 2019-11-21

**Authors:** Elaine Melo, Flavio Michels, Daniela Arakaki, Nayara Lima, Daniel Gonçalves, Leandro Cavalheiro, Lincoln Oliveira, Anderson Caires, Priscila Hiane, Valter Nascimento

**Affiliations:** 1Group of Spectroscopy and Bioinformatics Applied Biodiversity and Health (GEBABS), Graduate Program on Health and Development in West Central Region, School of Medicine, Federal University of Mato Grosso do Sul, UFMS, Campo Grande 79070-900, Brazil; daniarakaki@gmail.com (D.A.); nayaralima.01@hotmail.com (N.L.); 2Physics Institute, Federal University of Mato Grosso do Sul, Campo Grande 79070-900, Brazil; flavio.michels@ufms.br (F.M.); andercaires@gmail.com (A.C.); 3Department of Chemistry, Minas Gerais State University, UEMG, Ituiutaba 31360-900, Brazil; daniel.goncalves@uemg.br; 4Chemistry Institute, Federal University of Mato Grosso do Sul, Campo Grande 79070-900, Brazil; lernfc@gmail.com (L.C.); lincoln.cso@gmail.com (L.O.); 5Graduate Program on Health and Development in West Central Region, School of Medicine, Federal University of Mato Grosso do Sul, UFMS, Campo Grande 79070-900, Brazil; priscila.hiane@ufms.br

**Keywords:** babassu (*Attalea speciosa*) edible oil, cold press oil extraction, physicochemical characterization, oil stability and quality

## Abstract

Interest in edible oil extraction processes is growing interest because the final nutritional quality of the extracted oil depends on the procedure used to obtain ir. In this context, a domestic cold oil press machine is a valuable tool that avoids the use of chemicals during oil extraction, in an environmentally friendly way. Although babassu (*Attalea speciosa*) oil is economically important in several Brazilian regions due to its nutritional and healthy features, few studies have been conducted on the chemical composition and stability of babassu oils extracted by cold pressing. Babassu oil’s major constituents are saturated fatty acids (~86.42%), with the most prevalent fatty acids being lauric (~47.40%), myristic (15.64%), and oleic (~11.28%) acids, respectively, within the recommended range by Codex Alimentarius, presenting atherogenicity and thrombogenicity indexes favorable for human consumption. Peroxide value, Rancimat, and TGA/DSC results indicated that babassu oil is stable to oxidation. Also, macro- (Na, K, Ca, Mg, P) and micro-elements (Fe, Mn, Cr, Se, Al, and Zn) of babassu oil were determined, revealing levels below the tolerable upper intake level (ULs) for adults. These findings demonstrated that cold-press extraction using a domestic machine yielded a high-quality oil that kept oil chemical composition stable to oxidation with natural antioxidants.

## 1. Introduction

Daily intake of vegetable oils in adequate amounts may play an essential role in keeping the body healthy [1,2]. Due to the widespread use of edible oils in the culinary, food processing, and chemical industries, their consumption has increased considerably worldwide in recent years [3]. European Union, China, and India are the largest importers and consumers of edible oils [4].

In order to attend the health requirements demanded by consumers, in recent years, there has been a diversification in the cultivation of oilseeds. In addition to oils derived from oilseeds such as groundnut, soybean, sunflower, rapeseed mustard, cottonseed and so on [5], there was a large production of oils from various tree fruits, like coconut, palm, olive, sesame, etc., which provide higher yields [6]. Despite the wide variety of edible oils available on the market, information on their physical properties is still insufficient, making it challenging to choose nutritionally suitable oils for consumption [2] and appropriate for application in the food industry [7].

A critical indicator of the quality of edible oils is its oxidative stability, which is directly related to their shelf life [8,9,10]. Oxidative stability is influenced by the chemical composition of the oil and action of various factors with pro- and antioxidant characteristics, and it can occur during oil processing and storage [11]. Pro-oxidant factors include exposure to light, oxygen, heat, and the presence of trace elements in oils [1,12,13]. These factors can accelerate lipid oxidation, decrease oxidative stability and consequently cause significant impacts on sensory properties, nutritional depreciation, and decrease in the shelf life [11]. Besides, pro-oxidants can generate secondary oxidation products such as free fatty acids, peroxides, hydroperoxides, dienes, conjugated trienes, hydroxides, and ketones that can cause toxicity and contribute to the development of heart disease, cancer and atherosclerosis [10,14].

Brazilian Cerrado regions have a huge variety of plant species and are conducive to the growth and development of palm trees such as babassu, which can provide great nutritional resources and financial return for the poorer population [15]. Babassu also grows in the western portion of South America. Babassu is scientifically known as *Attalea speciosa* mart. Ex spreng, sinonym *Orbignya paleratha,* and *O. oleifera* [16], and its nuts are the second bestselling product in Brazil [16].

The production of babassu in Brazil is of great importance for agroextractivism and is mainly related to a large number of products and by-products derived from all parts of the plant (mesocarp, endocarp, epicarp, and leaves). Parts of babassu such as epicarp and endocarp are used in handicrafts, home coverings, burning in domestic and commercial ovens and organic manure. The uses for babassu oil extracted from nuts (endocarp) include human feed, cosmetics and fuel [15]. In addition, studies have shown the important activity of babassu oil microemulsion as a natural product to improve immune system function [17].

A recent study demonstrated that the physicochemical properties and fatty acid composition of babassu and indaiá (*A. dubia*) are similar in their composition of lauric acid, myristic acid, caprylic acid, and capric acid when extracted by a solvent extraction method [18]. However, studies on the mineral composition of various edible oils are scarce, especially the presence of macro and micro-elements in babassu oil, which may be influenced by agronomic, climatic and industrial applications [19].

To date, despite the fact athe physicochemical characterization of babassu oils has been performed, especially for determining the fatty acid composition of babassu oil obtained from Brazilian Amazon region [20], to the best of our knowledge, there is no available data on the macro- and microelements composition in babassu oil extracted by the cold press method. However, it is important to point out that Naozuka et al. [19] have demonstrated that there are metals, metalloids, and non-metals in the babassu nut and mesocarp. We believe this presence of metals in nuts indicates that babassu oil may become contaminated with metals during the production process. In addition, there are few studies on the properties and content of babassu oil extracted from other Brazilian regions, especially the obtained from the Brazilian Cerrado located in the Midwest region of the country.

Domestic demand for healthier vegetable oils and fats has been rising rapidly. It is extensively known that chemical contaminants have negative effects on the oxidative stability of oils and human health. To exploit this opportunity, companies have been producing domestic cold extraction machines for oil collection. The cold-press extracted oil is safe for consumers and does not involve either heat or chemical extraction, resulting in high-quality oils [21]. In fact, in Brazil and other countries, it has become common to use domestic cold extraction machines for extracting sunflower seed oil, flaxseed oil, and soybean oil.

Motivated by the manuscript published by Serra et al. [20], which emphasizes the need for new ways of oil extraction and filtration methods, the present investigation aimed to evaluate for the first time the physicochemical properties, thermal and oxidative stability, optical properties and mineral composition of edible babassu oil collected and consumed in the Midwest region of Brazil and extracted by cold pressing using a domestic extraction machine. Original results on thermal analysis, molecular absorbance (UV-VIS), and fluorescence, as well as quantification of macro- and micro-elements in babassu oil from the Midwest region of Brazil are presented in this study. Also, we compared the results of the fatty acids, and physicochemical profile with the values established by Codex Alimentarius [22] for vegetable oils. Furthermore, the mineral content obtained from babassu oil was compared to Codex Alimentarius values for refined oils and tolerable higher intake levels for adults (31 to 50 years of age) [23].

## 2. Results and Discussion

### 2.1. Sample Moisture and Yield

The determination of moisture in seeds of babassu revealed that it contains 4.153 ± 0.030% of water. Low amounts of water content usually yield a larger amount of oil extraction, which can be explained not only by the chemical composition but as well because the water may act as a lubricant, decreasing the pressure in the compression area of the press [2]. Oil yield was 5.6%, which means 265 mL from the starter material of 5 kg. It is worth to note that the yield can be increased by repressing the oil mass, which we did not do, since the primary goal of this paper was to evaluate the quality of the oil.

### 2.2. Fatty Acid Composition

In our study, 21 fatty acids were identified (Table 1). The fatty acids composition in the babassu oils decreases in the order: lauric (47.40%) > myristic (15.64%) > oleic (11.28%) > palmitic (8.01%) > caprylic (6.21%) > capric (5.78%) > stearic (3.15%) > linoleic (1.85%) > linolenic (0.25%) > elaidic (0.09%) > butyric (0.05%), arachidic (0.05%), gondoic (0.05%) > dihomo-γ-linolenic (0.04%), lignoceric (0.04%) > tridecanoic (0.03%) > undecilic (0.02%), palmitoleic (0.02%), margaric (0.02%) > behenic (0.01%) and cervonic (0.01%). Lauric acid (C12:0) was the predominant saturated fatty acid (SFA) comprising 47.40% of total fatty acid composition. Lauric oils and their derivatives have many applications in both the food and chemical industries. In addition, observed contents of myristic, palmitic and capric acid obtained from cold-pressed babassu oil using a domestic extraction machine was higher than those values of babassu oil using the artisanal cold pressing method using hydraulic presses [20].

Table 1 presents a comparison between the obtained values of fatty acid concentrations of babassu oil and the required values by the Codex Alimentarius, revealing that the cold pressing extraction machine can be used to extract high-quality oil as the extracted oil meets the requirements established by Codex Alimentarius [22] for babassu oil. However, for butyric, undecylic, tridecanoic, and elaidic acid values have not yet established by Codex Alimentarius [22]. Table 1 also shows that the babassu oil is majorly composed of SFA (86.42%), followed by monounsaturated fatty acids (MUFA, 11.43%) and the least was seen in the polyunsaturated fatty acids (PUFA, 2.15%). Serra et al. [20] have reported that babassu oil obtained from the northern region of Brazil presented 89.5, 9.0 and 1.0% SFA, MUFA and PUFA, respectively. After comparison, babassu oil from the Midwest region of Brazil with those from the Amazon studied by Serra et al. [20], we found a percentage difference between saturated fatty acids, monounsaturated fatty acids and polyunsaturated fatty acids of 3.56%, 21.25%, and 46.51%, respectively. Therefore, the obtained values in both studies do not show a pattern since fatty acids are dependent on temperature, soil, and climate [24]. In the northeast region of Brazil, the study by Santos et al. [25] shows babassu oils obtained from the same territorial range have similar chemical composition, regardless of the biome. However, the fatty acid composition results obtained by Santos et al. [25] are different from our results and those obtained by Serra et al. [20].

### 2.3. Atherogenicity and Thrombogenicity Index

Table 1 presents the nutritional quality indexes of babassu oil, which determined values are 8.72 and 3.63 for the atherogenicity index (AI) and thrombogenicity index (TI). In addition, the atherogenic and thrombogenic indexes in babassu oil are lower than the values for coconut oil (AI = 24.04 and TI = 10.90, respectively) [26]. However, AI and IT for cold-pressed oil using a domestic extraction machine are within the range obtained for Amazon nut oil blends (AI = 0.1–14.6 and TI = 0. 18–6.69) [7]. The high nutritional value of babassu oil is primarily related to its saturated fatty acids (SFA) profile [7], suggesting that the consumption of these oils could be benefiting to human health.

The effects caused by the consumption of saturated fatty acids are related to weight gain, lipid profile, inflammatory biomarkers, and coronary artery disease [26]. However, there is no consensus on a safe dose recommended for consumption on a normolipid diet that can generate only health benefits or harm [27]. The impact of health problems due to food in humans is significant, affecting all. Positive and negative effects of food depend on the daily dose and rate at which they are consumed [26].

According to Houston [28], the correlation between the intake of saturated fatty acids and the risk of coronary artery disease is complex and involves variables such as the source of SFA, number of SFA carbons, the biological properties of each SFA and the set of SFA in the diet. In fact, interventional and epidemiological studies on the cause and effect relationship due to saturated fatty acid consumption and the risk of coronary artery disease corroborate with the complexity proposition [29] established by Houston [28].

Fatty acids such as saturated, monounsaturated, and polyunsaturated acids have specific roles in different metabolic pathways, being beneficial or causing health risks depending mainly on the amount and frequency with which they are ingested in the diet [30]. Indeed, further studies on the role of SFA are required as well as their involvement in dyslipidemia [31].

In spite the consumption of SFA can adversely affect the health status, the use of SFA for cooking seems to be advantageous in comparison to PUFAs, which are prone to thermal-induced oxidation in elevated temperatures leading to development of hydroperoxides and subsequentially to aldehydes, which are toxic, while SFA confers thermal stability, generating little lipid oxidation products creating few health hazards [32].

### 2.4. Chemical Physical Analysis

The physicochemical aspects of oils represent an important tool for determining identity, quality, oxidative stability [7,20,31] and authenticity [22]. Table 2 shows the results of the physicochemical profile of babassu oil obtained in our study compared with those values established by Codex Alimentarius for cold-pressed oils and crude oil of babassu [22].

Peroxide index is used as a parameter for oil quality assessment [7], because it is one of the compounds produced during the beginning of the oxidation process [33]. Table 2 shows that the peroxide index was 2.40 mEq/kg and is below the Codex Alimentarius values (15 mEq/kg) for cold-pressed oil [22], which reinforces the oxidation stability quality desired for SFA content oils. On the other hand, it is higher than that found by Serra et al. [20] who found 0.43 mEq/kg of peroxide in cold-pressed babassu oil, and similar to that found in mururu (2.26 mEq/kg) and tucumã oils (2.14 mEq/kg) by Pereira et al. [7].

Results of the levels of peroxide found in babassu oil indicate that this oil may be stable to oxidation. A low peroxide level cannot be considered a good tool when used alone to assert the oxidative stability of oils, because, over time, the peroxide level behaves in a Gaussian manner [33,34]. Low peroxide content may occur due to storage time, contact with air and temperature variations [35]. Therefore, for the evaluation of oil quality, it is necessary to take into account the various organoleptic modifications that occur during the oil oxidation process and, consequently, to employ different methods for the evaluation of oxidative stability [33].

The acidity index represents the amount of free fatty acids after triglyceride hydrolysis [36]. Actually, the acidity index the most important physicochemical index used to evaluate oil quality [9]. The acidity index of babassu oil (3.47 mg KOH/g) is lower than the value stipulated by Codex Alimentarius [22] (4 mg KOH/g) (Table 2) and higher than found by Serra et al. [20] (1.06 mg KOH/g). Again, these results evidence the high stability of babassu oil.

The refractive index for oil extracted by cold pressing using a domestic extraction machine (1.46 °C) is within of values established by Codex Alimentarius for crude oil babassu (1.448–1.451 °C) and obtained by Serra et al. (1.47 °C) [20].

Iodine index measures the degree of oil unsaturation [9], i.e., the higher the iodine index, the greater the amount of unsaturated fatty acids and the higher the susceptibility to oxidation [7]. Iodine Index in babassu oil was 14.0 g iodine/100 g. The results reveal that babassu oil has a low degree of unsaturation and meets the requirements stipulated by Codex Alimentarius (10.0–18.0 g iodine/100 g) [22]. According to Serra et al. [20], low edible oil iodine values are associated with low lipid oxidation and good quality. Our result is consistent with that reported by Dijkstra [37], where the high lauric acid content reflects lower iodine value.

The saponification index represents the average molecular weight of fatty acids present in oil [35]. Thus, a high saponification index reflects the presence of short and medium-chain acids [20]. Babassu oil had a high saponification index (265 mg KOH/g) compared to the Codex Alimentarius [22] value (256 mg KOH/g) for crude oils and values found by Serra et al. [20] for babassu oil (249.50 mg KOH/g). On the other hand, the babassu oil saponification index is in agreement with the studies of Dijkstra [37], which suggests that oils rich in lauric acid, such as babassu, have a saponification value ranging from 245 to 265 mg KOH/g.

Unsaponifiable matter of babassu oil (0.40%) quantified in our study is close to oil from other palm trees such as tucumã (0.47%) [7] and Brazil nuts (0.45%). The unsaponifiable matter of oil consists of a mixture of bioactive compounds such as sterols, hydrocarbons, and pigments [38]. Therefore, due to the low value of unsaponifiable matter found in babassu, it is suggested the low concentration of the pro-oxidant compounds, which contributes to delay in the oil rancidification process and increases its stability.

### 2.5. Oxidative Stability Assessment by the Rancimat Method

Babassu oil induction period data obtained using Rancimat at 110 °C is shown in Figure 1. Babassu oil presented an oxidation induction period (33.69 h) higher than that found for palm oil (15.5 ± 1.5 h), sunflower (5.5 ± 0.5 h) oil, and rapeseed (4.5 ± 0.5 h) [39].

The evaluation of the accelerated oxidation induction period is widely used to monitor the lipid oxidation process of edible oils [40]. Results obtained in this study show high oxidative stability of babassu nut oil, which can be due to the high degree of saturation (86.42%) of the oil, among other factors, especially lauric fatty acid (47.40%). It is recommended to reduce oxidation, promoting factors such as exposure to light, oxygen and, heat to increase the oxidative stability of oils, as well as to intensify the antioxidant content in the oil [12]. The main factor that limits the shelf life of food is lipid oxidation [11].

### 2.6. Thermogravimetry/Derivative Thermogravimetry (TG/DTG)

TG and DTG curves of babassu oil in the presence of synthetic air (Air) and nitrogen (N_2_) atmospheres are shown in Figure 2. As can be seen from TG/DTG curves in Air, the first step of decomposition occurs between c.a. 203 to 396 °C. The thermal stability in this condition, by onset temperature (T_onset_), is 264.3 °C, and the first DTG peaks max occurs at 291 °C, respectively. The second mass loss step occurs from 396 to 576 °C. Considering the temperature range between c.a. 203 to 576 °C, there were mass losses of 94.2% and 5.6% which can be attributed to the term oxidative decomposition of lower molecular weight fatty acids and later of higher molecular weight acids and 0.33% residual carbonaceous formation. Under nitrogen (N_2_) atmosphere, a mass loss in the temperature range of c.a. 207 °C to 460 °C is observed. The thermal stability by onset temperature (T_onset_) is 289.4 °C, and the first DTG peaks max occur at 336.2 °C, respectively. The first step of decomposition, endset at temperatures of 364 °C with a mass loss of the 96.4%. For the second step, with a temperature endset of 460 °C, there was a loss mass of 3.07%. Mass losses are attributed to thermodegradation or volatilization of fatty acids and accumulation of 0.49% of the carbonaceous residue.

Aiming to clarify the steps of decomposition, the results of the analysis of thermal decomposition of babassu oil in synthetic air and nitrogen atmosphere under dynamic and almost isothermal conditions using TG and DTG are presented in Table 3. Figure 3 shows the TG/DTG curves of babassu oil in a quasi-isothermal mode of analysis on heating in nitrogen (N_2_) and synthetic air atmospheres.

In the quasi-isothermal analysis mode for babassu oil in the presence of synthetic air, there were three steps of mass loss, as following described. As can be seen from TG curve, the first step of decomposition occurs at the T_onset_ of 255.68 °C, in which there was a mass loss of 80.0%, attributed to the oxidative decomposition of saturated fatty acids with the lowest number of carbon atoms (6 at 16 C). The second mass loss step (13.0%) occurs from 255.60 to 409 °C. In the third stage, a mass loss of 6.9% in the initial temperature of the 409.26 °C and endset of the 555.22 °C occurred. Therefore, the losses in stages 2 and 3 happen when higher molecular weight and unsaturated fatty acids (18 C) degrades, as well as the formation of carbonaceous residues (0.20%).

Babassu oil curves in the quasi-isothermal mode of nitrogen (N_2_) heating analysis show that the first step of decomposition occurs at the T_onset_ of 263.57 °C with a mass loss of 87.7% due to thermodegradation of saturated fatty acids. In the second step for the temperature range 263.57 to 447 °C, the mass loss (11.5%) is attributed to the unsaturated fatty acids (18 C) with the formation of carbonaceous residues (0.80%).

In the present study, the onset temperature of cold pressing babassu oil in the atmosphere of synthetic air (264.3 °C) is lower than those of commercial oils, such as corn oil (306 °C), followed, by the sunflower one (304 °C) and olive oil (288 °C) [41]. Commercial oils have greater stability due to the refinement process that eliminates some pro-oxidant components [42], and the addition of synthetic antioxidants [43]. In fact, the process of refining of oils promotes the reduction in the total content of phenol, β-carotene, and oxygen radical absorbance [44].

Figure 4 shows the DSC babassu oil cooling curves for a temperature range from 60 to −60 °C and heating rate of 10 °C/min in an atmosphere of nitrogen (N_2_) with a flow rate of 50 mL min^−1^. As Figure 4 shows, the curve has two consecutive exothermic crystallization peaks, which may be due to polymorphic changes. The temperature onset point (T_onset_) starting at 5.0 °C, with a total energy of 79.9 J/g. According to Zhang et al. [45] and Tan et al. [46], the crystallization profile in oils are characterized by the beginning of fat crystal formation, which is related and greatly influenced by mass transfer, heat transfer, cooling rate, viscosity, presence of shear, etc. [47]. Zhang et al. [45] demonstrated that the isothermal photomicrographs of crystals formed from Palm Oil on various temperatures. Actually, the crystallization of fatty acids is a rearrangement of molecules due to the presence of saturated triglycerides.

According to the DSC analyzes for sunflower oil published by van Watten et al. [48], the crystallization temperature for commercial sunflower oil occurs from −30 to −35 °C at a rate of −10 °C/s. Thus, crystallization temperature in the babassu oil obtained in the present study was higher than the crystallization temperature of commercial sunflower oil.

The process of crystallization behaviors in oils is complex. Endothermic peaks or broad endotherms at temperatures below −30 °C in the cooling curve are attributed to the formation of freezing crystals commonly associated with unsaturated fatty acid esters. On the other hand, endothermic peaks or endotherms at a temperature between −10 and 15 °C are attributed to freezing crystal formation, which mainly consists of saturated fatty acid esters.

### 2.7. Molecular Absorption (UV-VIS) and Molecular Fluorescence Spectroscopy

Figure 5 shows the UV-VIS absorbance spectrum of babassu oil, with high absorption in the spectral region between 270 and 300 nm and the second with absorption at about 350–450 nm. The first range included the presence of unsaturated fatty acids including oleic and linoleic acids, whose peak around 280 nm was much more intense than those of the other components [49]. Besides, the second spectral ranges observed between 350 and 450 nm, as presented in the inset of Figure 5 corresponds to chlorophylls or carotenoids [50,51].

Recent studies have demonstrated that the molecular absorption bands of vegetable oils, in the UV-Vis range, can be used as a quality parameter for monitoring the oil oxidation [52,53]. The typical excitation-emission fluorescence map of babassu oils is shown in Figure 6. The fluorescence depends on sample concentration; therefore, spectra of the oil diluted at 1 × 10^−3^ g/mL and 0.05 g/mL are presented. The total fluorescence spectrum of babassu oils (Figure 6) exhibits two intense bands, one with excitation at about 275–300 nm and emission at about 300–325 nm and the second with excitation at about 300–325 nm and emission at about 425–525 nm, Figure 6. These fluorescence bands may be associated with endogenous antioxidants present in babassu vegetable oil, i.e., the presence of tocopherols and/or carotenoids [51].

Therefore, we once more conclude that the fluorescence emission from 275 to 400 nm has contributions from both tocopherols and phenolic compounds [54]. The second range with excitation length about 300–330 nm and emission at about 425–525 nm corresponds to carotenoids [50]. However, the emission observed in the λ = 425–525 nm range can also be correlated to the range of fatty acid oxidation products [49]. The fatty acid oxidation products and tocopherol present in blended oils is responsible for the fluorescence spectra in the emission from 400 to 500 nm [55].

Unlike commercially extracted oils and considering parameters such as safety, time, and low risk of contamination, babassu oil obtained by cold pressing extraction using a domestic extraction machine retains its healthy qualities, such as the presence of natural antioxidants.

### 2.8. Determination of Macro- and Micro-Elements

Table 4 shows the elemental concentration of macro- and micro-element data obtained from babassu oil (*A. speciosa*) compared to Codex Alimentarius values for refined oils and tolerable higher intake levels for adults (31 to 50 years of age). The relative standard derivations were less than 12% for all investigated elements. Results from spike and recovery experiments were in the range 90–103% (Table 5) for almost all the elements. Table 6 contains the operational conditions used in the ICP-OES analysis as wavelengths, detection limits (LOD), quantification (LOQ), and correlation coefficient (R^2^) for each element quantified in the present study.

The presence of metals in edible oils may be due to endogenous factors to which the plant is exposed, such as soil type, environment, use of fertilizers and pesticides, as well as exogenous sources that contaminated the oil during technique of processing such as extracting, crushing, refining, bleaching, hydrogenation and deodorisation [1,56]. However, in our study, the nuts of babassu were driven directly to the domestic extraction machine to avoid any form of contamination (see methodology).

It is well known that analysis of the elemental composition of vegetable oils can be used to monitor their quality, adulteration, and conservation of their products. For example, the determination of Ag, As, Ba, Be, Cd, Co, Cr, Cu Fe, Hg, Mn, Mo, Ni, Pb, Sb, Ti, Tl, and V has been used to evaluate the quality of virgin olive, olive, pomace-olive, sunflower, soybean and corn oils in Spain [1]; In Iran, the content of Pb, Cd, Ni, Mn, Zn, Cu, Fe, Ca and Mg were investigated to analyze olive, canola, sunflower and soybean oils [57]. Levels of Cu, Zn, Fe, Mn, Cd, Ni, Pb, and as were also used for monitoring edible oils, such as soybean, corn, peanut, sesame, rapeseed, cottonseed, olive, blend and sunflower oils in China [13]. In Turkey, olive, hazelnut, sunflower oils as well as margarine and butter were tested by means of the presence of Fe, Mn, Zn, Cu, Pb, Co, Cd, Na, K, Ca and Mg [58] while Pb, Cd, Ni and As has been quantified for quality control of olive oils produced in Italy [59]. In summary, these investigations have demonstrated that the potential health risks and/or health benefits for human consumption can correlate with the presence of macro- and microelement in edible oils. The elemental content in edible oils is useful in determining the quality of refined oils when comparing the values set by Codex Alimentarius [22] and also compared to tolerable upper intake levels for adults [23].

In the present study (Table 4), the concentration of macro-elements in the babassu oil decrease in the following order: Na (24.35 ± 2.780 mg/100 g) > P (12.75 ± 0.440 mg/100 g) > Ca (4.10 ± 0.06 mg/100 g) > Mg (2.25 ± 0.014 mg/100 g) > K (1.10 ± 0.012 mg/100 g), while concentration of micro-elements decrease in the order: Al (1.03 ± 0.003 mg/100 g) > Zn (0.45 ± 0.054 mg/100 g) > Cr (0.36 ± 0.011 mg/100 g) > Fe (0.13 ± 0.002 mg/100 g) > Mn (0.13 ± 0.002 mg/100 g) > Se (0.09 ± 0.008 mg/100 g). Elements such as copper, nickel, cobalt, cadmium and molybdenum had a concentration below the detection limit (see Table 4). In addition, for vegetable oils, the Codex Alimentarius limit have not established for Na, K, Mg, Mn, Ni, Co, Cr, Se, Al, Cd, Mo and Zn.

Table 4 presents that the iron concentration obtained was lower than the maximum value recommended by Codex Alimentarius (0.15 mg/100 g) [22]. Iron and copper present in the oil may react directly with the lipids and produce lipid alkyl radicals and reactive oxygen species that decrease the oxidative stability of the oil and accelerate the oxidation process [10]. In general, the oxidative stability of the oil is altered due to the presence of pro-oxidant components such as trace elements that facilitate oxidative degradation and contribute to the emergence of adverse effects such as color, odor, and flavor alteration. Also, trace elements reduce product shelf life and alter the nutritional value of the oils [57].

According to Naozuka et al. [19], there are metals (Al, Ba, Ca, Cu, Fe, K, Mg, Mn, Sr, and Zn), metalloids (B and Si), and non-metals (Cl, P, and S) in the babassu nut and mesocarp. Thus, the babassu oils obtained in our study contain various chemical elements in their composition that must come from nuts. We compared the results obtained within the framework of this study (Table 4) to the tolerable upper intake levels (ULs) for adults (31–50 y) estimates based on dietary reference intake (DRI) committees of Food and Nutrition Board of the Institute of Medicine. It is very important to determine the concentration of macro- and micro-elements in edible oil to know if these concentrations are above the limit tolerable upper intake levels (mg/day), that is, maximum daily intake of nutrients that can pose a risk of adverse effects on health. After comparison, the concentration of macro- and micro-elements with those proposed by ULs in the cold extraction babassu oil in a homemade machine model, it is found these content in babassu oil are below the values tolerable upper intake level. Consequently, the present study demonstrated that the babassu oil does not pose a health hazard risk regarding elemental concentration [23], being safe for human consumption.

Table 6 shows the limit of LOD, LOQ, correlation coefficient (R^2^), and analytical wavelength. The detection limit (LOD) and quantification (LOQ) values for the 16 elements analyzed by ICP OES had a range of 0.0001–0.03 and 0.0005–0.1, respectively. The correlation coefficient for the calibration curves ranged from 0.9993 to copper and 0.9999 to K, Mg, P, Se, and Mo.

## 3. Materials and Methods

### 3.1. Fruit Collection, Seed Moisture and Oil Production

We collected several ripe babassu fruits that were lying on the ground in Campo Grande, Mato Grosso do Sul state, Brazil, in September 2018 (Figure 7a). The harvest of babassu fruit selection was at random in three different sites within the State of Mato Grosso do Sul (MS). Initially, the samples were prepared for analysis in the research laboratory of Spectroscopy and Bioinformatics Applied Biodiversity and Health, School of Medicine, Federal University of Mato Grosso do Sul. After collection, the almonds were separated from the fruits (Figure 7b) and conducted for oil extraction and water content.

For moisture determination, we homogenized and weighted 5 g of babassu seeds, and kept in oven at 105 °C, being hourly weighted after rest in desiccator until achievement of constant weight.

The oils of the nuts (Figure 7c) were obtained through methods by cold pressing using a Stainless Steel Worm Thread Machine (Yoda Nut & Seed Cold Press Oil Extractor-Gourmet Extractor, oil Natural, Homeup, (Yoda Europe, Cluj-Napoca, Romania).

The oil extraction machine has an internal filter, so the oil obtained is already filtered and ready for analysis. A representative sample of 265 mL was obtained from batch (5 Kg) of the oils of fruits. Aliquots of oils were stored in amber glass vials and protected from light and kept at temperatures of 10 °C until analysis.

### 3.2. Fatty Acid Composition

The fatty acid compositions of the oils were identified according to the methodology proposed by Bligh and Dyer [60], which considered the proportionality recommendations between solvents, methanol, chloroform, and water (2:1:0.8). The obtained fatty acid methyl esters derivatives (FAMEs) were prepared from samples by saponified with methanolic NaOH and then esterified with a mixture of H_2_SO_4_ and NH_4_Cl in methanol and transferred to hexane according to [61,62].

FAMEs were analyzed using a gas chromatograph (model CP-3800, Varian, Santa Clara, CA, USA) equipped with flame ionization detector, a split/splitless injector, and stationary phase fused silica capillary column of polyethylene glycol (Carbowax 20 M, Length 30 m × 0.25 mm, Quadrex, Santa Clara, CA, USA). We used the following operational parameters for chromatography: the injector and detector temperatures were 250 °C; The column temperature was programmed to 80 °C for 2 min, followed by a ramp of 4 °C/min up to 220 °C and kept for 13 min; hydrogen carrier gas with 1 mL/min flow and injection volume 1 μL. Retention times were compared with the respective methyl ester standards (Supelco, F.A.M.E. mix C4:0 to C24:0, Sigma-Aldrich, Darmstadt, DA, Germany).

### 3.3. Index of Atherogenicity (IA) and Thrombogenicity (IT)

The evaluation of the nutritional quality of babassu oil based on its fatty acid composition was determined by the index of atherogenicity (Equation (1)) and thrombogenicity (Equation (2)), as proposed by [7]. Equations (1) and (2) used to calculate the *IA* and *IT* consider the different effects of different fatty acids on human health:(1)$$IA=\frac{C12:0+C14:0+C16:0}{\sum MUFA+\sum w^{3}+\sum w^{6}}$$
(2)$$IT=\frac{C14:0+C16:0+C18:0}{\left(0.5\;x\sum MUFA\right)\;+\left(3\;x\;\sum \omega 3\right)+\left(0.5\;x\;\sum \omega 6\right)}$$

The terms in Equations (1) and (2) refer to lauric acid (C12:0), myristic acid (C14:0); palmitic acid (C16:0); stearic acid (C18:0); monounsaturated fatty acid (MUFA); omega-3 fatty acid (ω3) and omega-6 fatty acid (ω6).

### 3.4. Evaluation of the Identity and Quality Characteristics of Oils Samples

The characterization of babassu oil was performed according to American Oil Chemist’s Society [63] methodology for quality parameters, in triplicate, namely: peroxide index (Cd 8-53) and acidity index (Ca 5a-40); and identity parameters: iodine index-Wijs method (Cd 1-25), saponification index (Cd 3-25), unsaponifiable matter (Ca 6a-40) and refractive index (Cc 7-25).

### 3.5. Determination of the Oxidative Stability of Oil

Accelerated oxidation test was determined by the Rancimat model 893 method (Metrohm Co, Basel, Switzerland) according to the European Union standardized standard EN 14112. The oxidation induction period (PI, hours) was determined when 3.0 g of oil sample was exposed to a continuous flow rate (10 L/h) and a constant temperature of 110 °C.

### 3.6. Thermal Analysis: Thermogravimetry (TG), Derivative Thermogravimetry (DTG) and Differential Scanning Calorimetry (DSC)

TG/DTG curves were performed using TGA Q-50 equipment (TA Instruments, New Castle, DE, USA). Two types of TG/DTG curves were obtained. Approximately 4 mg of babassu oil was added to a platinum crucible under nitrogen or synthetic air atmospheres at a flow rate of 60 mL/min, with temperatures range between the ambient one to 700 °C. First in a dynamic sequence with heating rate of 10 °C min^−1^ and than an quasi-isothermal cyclic sequence with an initial dynamic step with heating rate of 10 °C min^−1^, followed by the isothermal step and finally a new dynamic step, conditioned by: After the start of this curve, if mass loss is bigger than 2%, the step is finished and follows up by an isothermal step of the 135 min, and then, if mass loss is smaller than 2%, this step is also finished, returning to a dynamic step up to 700 °C.

DSC analyses were performed at DSC-Q20 equipment, coupled to an RCS90 refrigeration system (TA Instruments). The DSC curves were obtained using approximately 4 mg of babassu oil in an aluminum crucibles and with reference a similar crucible empty under nitrogen atmosphere with a flow rate of 50 mL/min, heating/cooling rate of 10 °C/min in heating cycles and subsequently cooling in temperature ranges from −60 °C to 60 °C.

In this paper, we determined the thermal stability by the Onset Point Temperature (T_onset_). The obtaining of the Onset Point is by the extrapolated beginning of the curve, which is defined by the point of intersection of the tangent with the point of maximum slope, on the principal site of the mass loss curve with the baseline extrapolated (TA Advantage/Universal Analysis Software).

### 3.7. Molecular Absorption (UV-VIS) and Molecular Fluorescence Spectroscopy

Babassu oil was diluted in HPLC grade hexane at a concentration of 10 g/L and from stock solution, we prepared different dilutions for spectra reading at 1 × 10^−3^ g/mL and 0.05 g/mL. Absorbance in the UV-Visible range was measured using a spectrophotometer (Lambda 265UV-VIS, Perkin Elmer, Waltham, MA, USA). An optical cuvette made from quartz with a 10 mm optical path and with four optical sides was used as a sample holder. The collection of UV-Vis absorption spectra was in the 220–420 nm range. All analyses were performed at room temperature.

Excitation-emission matrix fluorescence spectra were measured using a spectrofluorometer (Cary Eclipse, Varian). The excitation-emission maps of fluorescence were obtained by exciting the samples in the wavelengths from 200 to 375 nm in 5 nm steps and collecting the emission between 250 and 500 nm in 1 nm steps. The excitation and emission slits were 5 nm, and the sensitivity of the detector was 600 V. The samples were diluted in HPLC grade hexane at a concentration of 10 g/L. A four-sided quartz cell with a 10 mm optical path was used.

### 3.8. Microwave-Assisted Digestion and Analysis by ICP OES

An amount of 0.7 g of oils was weighted directly into Teflon DAP60^®^ vessels, and 6 mL of HNO_3_ (65%, Merck, Darmstadt, Germany) and 2 mL of H_2_O_2_ (30%, Merck) were added. The oil samples digesting occurred in a microwave system (Speedwave four^®^, Berghof, Germany) on conditions reported in Table 7, respectively. The blank solutions were prepared in the same way as the samples, but without adding oil.

After completion of digestion, the solutions containing the samples were transferred to a polyethylene tube, and ultra-pure water (18 MΩcm, Milli-Q Millipore, Bedford, MA, USA) was added to complete a final volume of 30 mL. After that, the concentrations of the macro e microelements (Na, K, Ca, Mg, P, Fe, Mn, Ni, Cu, Co, Cr, Se Al, Cd, Mo, and Zn) in oil of the babassu was determined by technique of Inductively Coupled Plasma-Optical Emission Spectrometer (ICP OES) with an Axial Plasma (iCAP 6000 Series, Thermo Scientific, Cambridge, UK).

Standard solutions were prepared by diluting a standard multi-element stock solution (SpecSol, Quinlab, Jacarei, SP, Brazil) containing 1000 mg/L of each element. We used five different concentrations to build calibration curves for the quantitative analysis of oils. The concentration range for the elements was 0.01–5.0 mg/L. A recovery test was performed; the solutions were spiked with 1 ppm. The spiking solution was made from a single multielement stock solution of 1000 ppm. The setup of ICP OES instrumental conditions for analysis listed in Table 8.

The limits of detection (LOD) and quantification (LOQ) were calculated using the background equivalent concentration and signal-to-background ratio, according to Currie recommendations [64].

## 4. Conclusions

Babassu oil is composed of saturated and unsaturated fatty acids, embracing several groups of SFAs, MUFAs, and at a minor level, PUFAs. Lauric and myristic acid was found to be the predominant SFAs, playing a major role in contributing to babassu oil stability. The values of fatty acid composition, acidity index, peroxide index, saponification index of babassu oil extracted by cold pressing using a domestic extraction machine were above the artisanal methods by cold pressing using hydraulic presses. According to the atherogenicity and thrombogenicity indexes, the obtained results demonstrated that babassu oil is favorable for consumption. Results of the levels of peroxide and Rancimat method revealed that babassu oil is very stable to oxidation, which favors its use to cooking or using as a frying oil, once its stability minimizes the formation of hazardous products. These assumptions are also supported by the findings from TGA/DSC data on the stability of the babassu oil in different atmosphere and temperatures, besides providing information on the thermal events such as melting and crystallization determining accurate and precise transition temperatures. Differences in the content of total fatty acids, of saturated and of mono-unsaturated fatty acids of babassu oil can be explained by the type of soil and climate of where the seed was collected. The cold press method preserved the important features of crude oil, such as the presence of natural antioxidants (tocopherols, phenolic compounds, and carotenoids), contributing to the final product quality.

In addition, the macro- and microelements of babassu oil extracted by cold-pressing were quantified for the first time. The results showed that babassu oil contained levels of macro- (Na, K, Ca, Mg, P) and micro-elements (Fe, Mn, Cr, Se, Al, and Zn) below the tolerable upper intake levels (ULs) for adults. In fact, obtaining cold-pressed oils using a home extraction machine can provide a low concentration of contaminating metals in oils, avoiding contamination due to refinement processes and the use of chemical agents.

From these results, we may conclude that the obtained babassu oil can be used as oil for daily consumption. However, the values of the composition of babassu oil fatty acids from the Midwest region differ from those from the Amazon region. Babassu oil fatty acid composition is within international recommendations established by Codex Alimentarius.

The cold-pressed extraction of babassu oil using a domestic extraction machine presented a low-cost way to obtain a high-quality edible oil, stable to oxidation, with low concentrations of metals, yielding a final product able to keep its natural antioxidants and, consequently, a healthier oil than those oils extracted by solvents or hot press techniques.

Furthermore, the present investigation also demonstrated that babassu oil contains different macro- and micro-elements, such as Na, K, Mg, Mn, Ni, Co, Cr, Se, Al, Cd, Mo, and Zn, which is not yet established by Codex Alimentarius for crude oils.

Finally, it is worth to point out that the present findings improve the knowledge regarding the physicochemical characterization and nutritional content of babassu oil as well as reveal a few gaps to be filled in the quality control of edible oils. Additionally, the presented data may be useful for the food industry and assist in the development of new studies on cold-pressed edible oils.

## Figures and Tables

**Figure 1 molecules-24-04235-f001:**
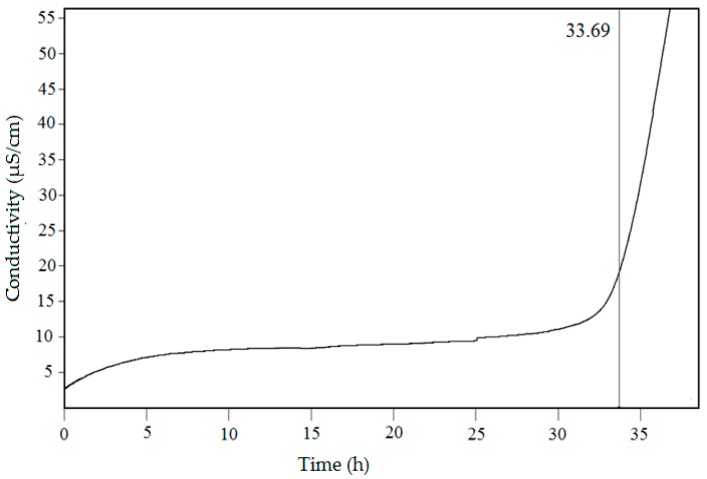
Conductivity versus time determined by the Rancimat method. Oxidation stability of oils at 110 °C. The induction time of babassu oil was 33.69 h.

**Figure 2 molecules-24-04235-f002:**
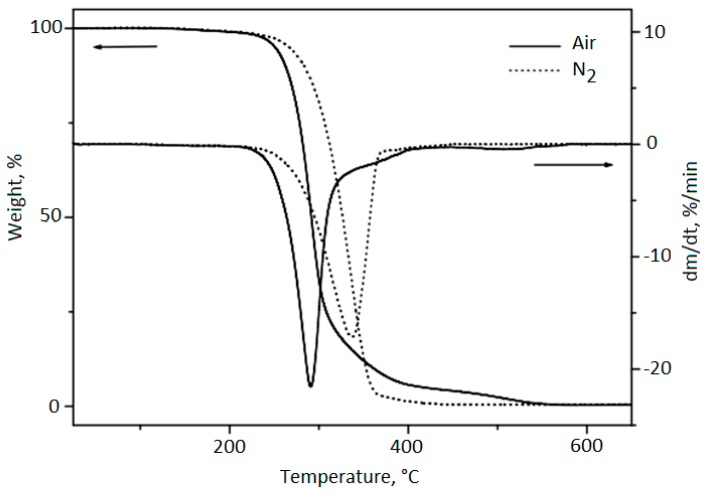
TG/DTG curve: Mass loss of babassu oil in an oxidative atmosphere of synthetic air and nitrogen under dynamic condition.

**Figure 3 molecules-24-04235-f003:**
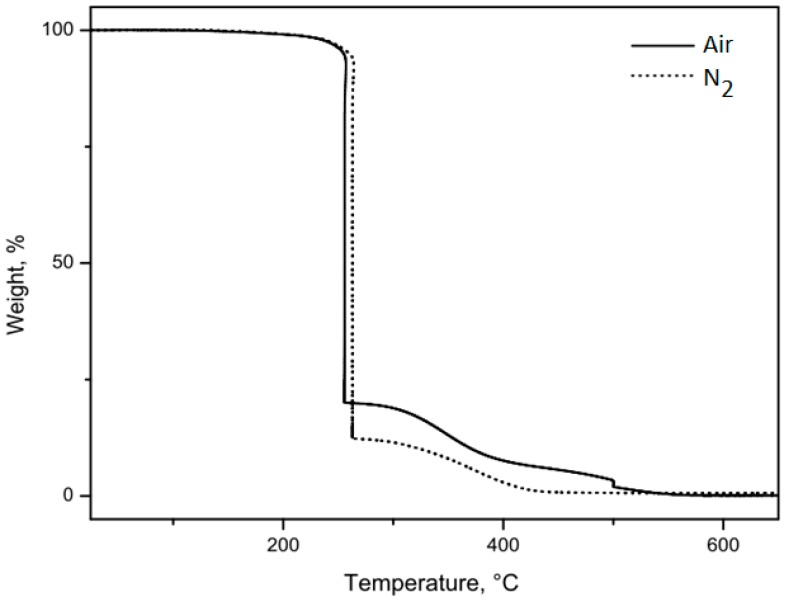
TG/DTG curve: Mass loss of babassu oil in an oxidative atmosphere of synthetic air and nitrogen under almost quasi-isothermal conditions.

**Figure 4 molecules-24-04235-f004:**
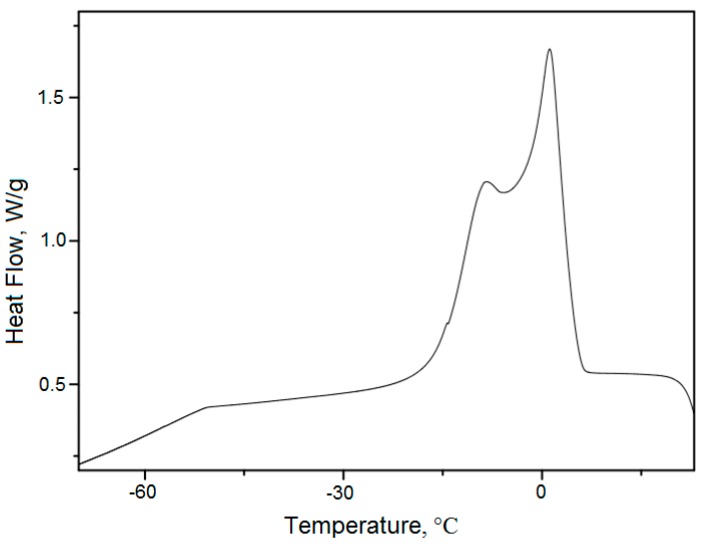
Cooling of babassu oil on an oxidative atmosphere of nitrogen under almost quasi-isothermal conditions.

**Figure 5 molecules-24-04235-f005:**
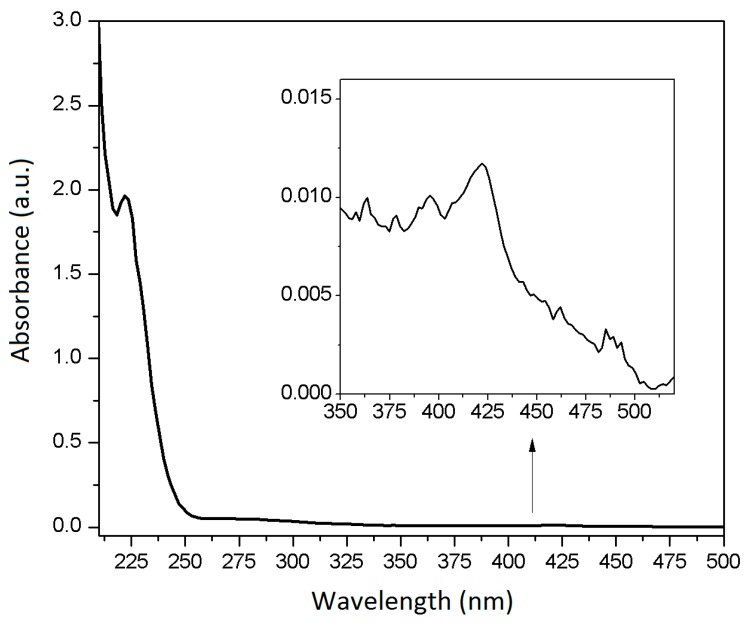
UV-VIS absorption spectrum (wavelength from 210 to 500 nm) of babassu oil diluted in hexane at 0.0100 g L^−1^). Inset: Zoom of the UV-Vis spectrum in the 350 to 500 nm range.

**Figure 6 molecules-24-04235-f006:**
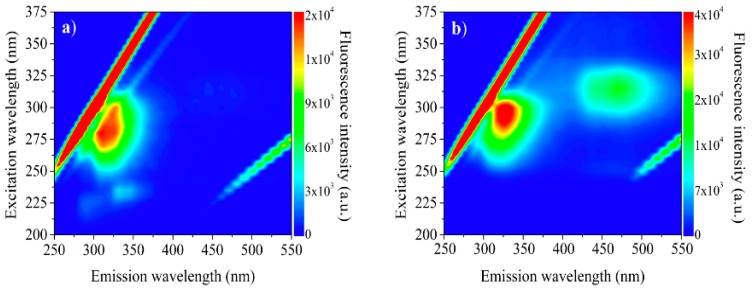
Excitation-emission map of oils from the babassu obtained by exciting between 200–375 nm and emission in the 250–550 nm range (concentration: (**a**) 1 × 10^−3^ and (**b**) 0.05 g/mL).

**Figure 7 molecules-24-04235-f007:**
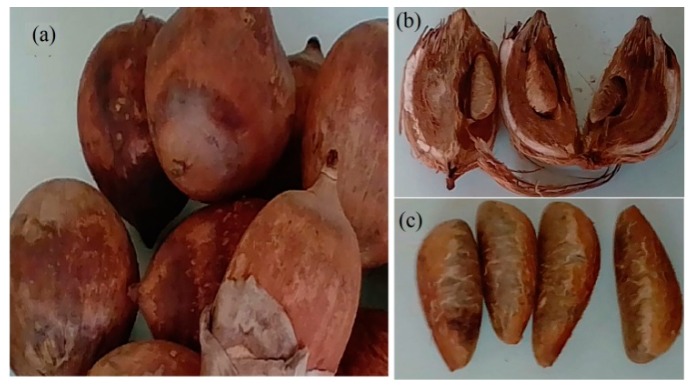
(**a**) Fruit, (**b**) endocarp and (**c**) nut of babassu fruits.

**Table 1 molecules-24-04235-t001:** Fatty acid profile of babassu oil obtained in this study compared to the Codex Alimentarius parameter for babassu oil.

Fatty Acids	Composition (%)	Babassu Oil Codex Alimentarius [22]
butyric (C4:0)	0.05	------
caprylic (C8:0)	6.21	2.6–7.3
capric (C10:0)	5.78	1.2–7.6
undecylic (C11:0)	0.02	------
lauric (C12:0)	47.40	40.0–55.0
tridecyl (C13:0)	0.03	------
myristic (C14:0)	15.64	11.0–27.0
palmitic (C16:0)	8.01	5.2–11.0
palmitoleic (C16:1)	0.02	ND
margaric (C17:0)	0.02	ND
stearic (C18:0)	3.15	1.8–7.4
oleic (C18:1n9)	11.28	9.0–20.0
elaidic (C18:1)	0.09	------
linoleic (C18:2)	1.85	1.4–6.6
linolenic (C18:3)	0.25	ND
arachidic (C20:0)	0.05	ND
gondoic (C20:1)	0.05	ND
dihomo-γ-linolenic (C20:3)	0.04	ND
behenic (C22:0)	0.01	ND
cervonic (C22:6)	0.01	ND
lignoceric (C24:0)	0.04	ND
ΣSFA	86.42	
ΣMUFA	11.43	
ΣPUFA	2.15	
atherogenicity index	8.72	
thrombogenicity index	3.63	

ND—non-detectable, defined as < 0.05%.

**Table 2 molecules-24-04235-t002:** Physicochemical aspects of babassu oil compared to quality parameters from Codex Alimentarius.

Parameters	Babassu Oil	Maximum Values [22]
Peroxide index (mEq/kg)	2.40	15 ^(a)^
Acidity Index (mg KOH/g)	3.47	4 ^(a)^
Refractive Index (°C)	1.46	1.448–1.451 ^(b)^
Iodine Index (g iodine/100 g)	14.0	10.0–18.0 ^(b)^
Saponification Index (mg KOH/g)	265	245–256 ^(b)^
Unsaponifiable Matter (%)	0.40	≤1.2 ^(b)^

^(a)^ Reference parameter for cold-pressed oils; ^(b)^ Reference value for crude oil of babassu.

**Table 3 molecules-24-04235-t003:** Results obtained from the TG/DTG curve of babassu oil in a synthetic air (Air) and nitrogen (N_2_) atmosphere under dynamic and almost isothermal conditions.

Curve	Sample	Steps	Temperature (°C)		T_onset_ (°C)	Δ Mass (%)
			Initial	End		
Dinamic	Babassu/Air	1°	203	396	264.3	94.2
		2°	396	576		5.6
	Babassu/N_2_	1°	207	364	289.4	96.4
		2°	364	460		3.7
Quasi-isothermal	Babassu/Air	1°	209	256	255.7	80.0
		2°	255	409		13.0
		3°	409	555		6.9
	Babassu/N_2_	1°	215	264	263.6	87.7
		2°	264	447		11.5

**Table 4 molecules-24-04235-t004:** Elemental concentration of macro and microelements in the oil of the babassu (*A. speciosa*) compared with values of the Codex Alimentarius for refined oils and Tolerable Upper Intake levels for adults (31–50 y).

Elements	Concentration (mg/100 g)	Codex Alimentarius [22] (mg/100 g)	Tolerable Upper Intake Levels (ULs, mg/day) [23]
Macro-elements			
Na	24.35 ± 2.780	NE	2300
K	1.10 ± 0.012	NE	ND
Ca	4.10 ± 0.06	NE	2500
Mg	2.25 ± 0.014	NE	350
P	12.75 ± 0.440	NE	4000
Micro-elements			
Fe	0.13 ± 0.002	(0.15)	45
Mn	0.13 ± 0.002	NE	11
Ni	<LOD	NE	1
Cu	<LOD	(0.01)	10
Co	<LOD	NE	ND
Cr	0.36 ± 0.011	NE	ND
Se	0.09 ± 0.008	NE	0.40
Al	1.03 ± 0.003	NE	ND
Cd	<LOD	NE	ND
Mo	<LOD	NE	2
Zn	0.45 ± 0.054	NE	40

NE—not established. ND—not determined.

**Table 5 molecules-24-04235-t005:** Recoveries for spiked babassu oil (%).

Element	Spike Recovery
Al	91
Ca	95
Co	92
Cr	103
Cu	98
Fe	90
K	95
Mg	91
Mn	99.8
Na	97
Ni	103
P	90
Se	98
Cd	90
Mo	98
Zn	92

**Table 6 molecules-24-04235-t006:** Operating conditions used in the analysis by ICP OES.

Elements	Wavelength (nm)	LOD (mg/L)	LOQ (mg/L)	Correlation
Na	589.592	0.0001	0.0005	0.9995
K	769.896	0.01	0.04	0.9999
Ca	422.673	0.02	0.08	0.9997
Mg	285.213	0.01	0.03	0.9999
P	185.942	0.01	0.04	0.9999
Fe	259.940	0.02	0.07	0.9996
Mn	257.610	0.001	0.004	0.9996
Ni	221.647	0.002	0.008	0.9996
Cu	327.396	0.002	0.006	0.9993
Co	238.892	0.006	0.02	0.9995
Cr	425.435	0.001	0.004	0.9994
Se	196.090	0.007	0.02	0.9999
Al	396.152	0.03	0.1	0.9997
Cd	228.802	0.0003	0.0009	0.9996
Mo	202.030	0.0006	0.0019	0.9999
Zn	213.856	0.0005	0.0046	0.9996

**Table 7 molecules-24-04235-t007:** Optimized program for the microwave digestion parameters for oil of the babassu.

Reagents			Volume	
HNO_3_ (65%)			6.0 mL	
H_2_O_2_ (35%)			2.0 mL	
**Step**	**Temperature (°C)**	**Pressure (bar)**	**Time (min)**	**Power (%)**
1	160	40	5	90
2	200	40	15	90
3	50	40	10	0
4	50	40	0	0
5	50	50	0	0

**Table 8 molecules-24-04235-t008:** Instrumental parameters for elemental determinations using axially viewed ICP OES.

Parameters	Setting
Power RF	1150 W
Plasma flow	12 L min^−1^
Sample flow rate	0.45 L min^−1^
Auxiliary flow	0.5 L min^−1^
Nebulizer	20 psi
Integration	15 time(s)
Stabilization	20 time(s)
Gas (99.999%)	Ar
Measure the analytical signal	3 replicates

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
