# Peer review of "First Study on the Oxidative Stability and Elemental Analysis of Babassu (Attalea speciosa) Edible Oil Produced in Brazil Using a Domestic Extraction Machine"

_molecules, 2019, doi:10.3390/molecules24234235_

Round 1

Reviewer 1 Report

The manuscript “First study on oxidative stability and elemental analysis of babassu (Attalea speciosa) edible oil produced in Brazil using a domestic extraction machine”, by Elaine Melo, Flavio Michels, Daniela Arakaki, Nayara Lima, Daniel Gonçalves, Leandro Cavalheiro, Lincoln Oliveira, Anderson Caires, Priscila Hiane and Valter Nascimento, presents interesting information on the composition of Brazilian babassu oil extracted by cold pressing. Certain characteristics like oxidative stability and thermal stability were also measured. This study was carried out in order to determine whether or not babassu oil can be used in human feed, therefore, quality parameters of Codex Alimentarius were considered for comparisons.

Concerning the structure of the manuscript, the authors tried to present the relevant background for their work in the introduction. However, some fragments of the text are difficult to understand likely because the authors did not consider the semantics of words during the redaction. This is an aspect that may seriously affect the quality of the manuscript. The English must be improved; although I have included some corrections in the PDF document and suggested the revision of some specific sentences (please see the attached pdf file), the authors must perform a careful revision of the text to make it more fluent and dynamic. It is highly recommended the revision by a native English speaker.

The description of materials and methods shows some deficiencies concerning the spectroscopic measurements.

I think that the manuscript can be improved by including some explanatory notes and additional references. Several fragments of the text should be rephrased. The following details should be corrected (considered) before the manuscript is accepted:

Please check the redaction of the sentence on page 1, lines 32 and 33. This sentence is not understood and a rephrase is needed. The sentence on page 1, lines 39 and 40 is not clear. Please rephrase it. On page 2 (line 61), the authors wrote that "oxidative stability is a "process". I don't think this is true, in any case, it is a chemical property. There are processes involved in the oxidation of the oil... this is different! On page 2 (lines 77 and 78), the authors wrote: "Several parts of babassu are used in the food human and animal,...". This expression seems incorrect. On page 11 (lines 347-352), it is unclear the description presented by the authors. The author should introduce the criteria they used to select the metals studied in this work and then present their own data. They can write: "Previous works on babassu oil were considered to select the metals measured in this work; Ag, As,... were measured in oil samples from Spain [1]; Pb, Cd,... were measured in oil samples from China [13];..." and so on. In Figure 5, I don't understand why the authors used four concentrations for UV/VIS spectroscopic measurements. It makes no sense since a high concentration implies a high absorption, and the aim was to find wavelength values in which the oil showed absorption of the radiation. Therefore, the measurement of one concentration would be enough. If the authors have arguments to justify the measurement of four oil dilutions, they must include explanatory notes in "Materials and Methods". By the way, in Materials and Methods, no mention of these four concentrations was made. On page 12 (lines 381 and 382), the sentence is wrong... nobody has ingested babassu oil for this study. Babassu oil contains macro- and micro-nutrients with levels below tolerable values considered for human feed. The authors must pay special attention to the redaction, otherwise they will confuse the readers.

In summary, I conclude that this paper requires a lot of work for its correction (especially in the redaction of the Introduction). Only after major revisions, this paper can be considered for publication in Molecules.

Author Response

Referee 1

All grammar suggestions over the text have been accepted, and we agree that the text needed improvement, so we thoroughly checked the text. Thank you so much for your suggestion. Yes, we have restructured the sentence for better understanding. 

Insights regarding sample concentration and UV-VIS spectra were clarified on page 11. Also, we provided further information on sample dilution on the method section (page 17).

The authors did not imply that anyone had consumed babassu oil on page 12, and the comment concerning that the oil is suitable for human feed is 1- regarding elemental content (as stated in the text); 2- after all the analysis regarding oil composition, stability, and the use of extraction method that excludes the use of solvents is only logical to achieve this conclusion. Also, on the 2nd page of the manuscript, we provided the information that the oil is proper for food human and animal (reference 15: Carrazza, L.R.; Silva, M.L.; Ávila, J.C.C. Manual Tecnológico de Aproveitamento Integral do Fruto do Babaçu. 2º ed.; Instituto Sociedade, População e Natureza (ISPN): Brasília, Brasil, 2012; pp. 01–63).

The manuscript has been revised as per the comments given by the reviewer, and our responses to all the comments are as follows:

Reviewer 1:  

Questions reviewer 1 : The English must be improved;

Answers to the reviewer:  Dear reviewer , the english text was corrected by a native although we only had ten days to correct it.

Questions reviewer 1 : the authors must perform a careful revision of the text to make it more fluent and dynamic. It is highly recommended the revision by a native English speaker.

Answers to the reviewer:  We believe that text after corrections has become more fluent and dynamic.

Questions reviewer 1 : The description of materials and methods shows some deficiencies concerning the spectroscopic measurements.

Answers to the reviewer:  Some parts of the text have been changed to make it easier to interpret.

Questions reviewer 1 : I think that the manuscript can be improved by including some explanatory notes and additional references. Several fragments of the text should be rephrased. The following details should be corrected (considered) before the manuscript is accepted:

Answers to the reviewer:  Several paragraphs have been rewritten according to their suggestions and doubts.

Questions reviewer 1 :

Answers to the reviewer:  In the next pages, all your suggestions were accepted “Please check the redaction of the sentence on page 1, lines 32 and 33. This sentence is not understood and a rephrase is needed. The sentence on page 1, lines 39 and 40 is not clear. Please rephrase it. On page 2 (line 61), the authors wrote that "oxidative stability is a "process". I don't think this is true, in any case, it is a chemical property. There are processes involved in the oxidation of the oil... this is different! On page 2 (lines 77 and 78), the authors wrote: "Several parts of babassu are used in the food human and animal,...". This expression seems incorrect. On page 11 (lines 347-352), it is unclear the description presented by the authors. The author should introduce the criteria they used to select the metals studied in this work and then present their own data. They can write: "Previous works on babassu oil were considered to select the metals measured in this work; Ag, As,... were measured in oil samples from Spain [1]; Pb, Cd,... were measured in oil samples from China [13];..." and so on. In Figure 5, I don't understand why the authors used four concentrations for UV/VIS spectroscopic measurements. It makes no sense since a high concentration implies a high absorption, and the aim was to find wavelength values in which the oil showed absorption of the radiation. Therefore, the measurement of one concentration would be enough. If the authors have arguments to justify the measurement of four oil dilutions, they must include explanatory notes in "Materials and Methods". By the way, in Materials and Methods, no mention of these four concentrations was made. On page 12 (lines 381 and 382), the sentence is wrong... nobody has ingested babassu oil for this study. Babassu oil contains macro- and micro-nutrients with levels below tolerable values considered for human feed. The authors must pay special attention to the redaction, otherwise they will confuse the readers.”

_________________________________________________________________

Page 1, line 28-48

Is written:

Abstract: People are becoming more interested in healthier edible oils and natural foods. This is where the domestic oil press machine is becoming a valuable tool. Babassu oils are important for human nutrition and economics in several Brazilian regions. Nevertheless, few studies have been conducted on the composition of babassu oils extracted by cold pressing. The aim of this manuscript was to study the physicochemical properties, oxidative stability, optical properties, thermal stability and mineral composition of babassu edible oil consumed in the Midwest region of Brazil and extracted by cold pressing using a domestic extraction machine. Babassu oils is comprised mainly by Saturated Fatty acids, (86.42 %), followed by monounsaturated fatty acids (MUFAs), (11.43%) and Polyunsaturated fatty acids (PUFAs) (2.15%). Lauric (47.40%), myristic (15.64%), oleic (11.28%) acids are the major fatty acids in the oil composition, with values within form those proposed international recommendations established by Codex Alimentarius. Index of atherogenicity and index of thrombogenicity shows babassu oils are favorable for consumption. Results of the levels of peroxide, Rancimat method and TGA/DSC in babassu oil indicate that this oil is stable to oxidation. According to absorbance and fluorecence results, babassu oils contain fluorophores as tocopherols, phenolic compounds, and chlorophylls. Babassu oil contained levels of macroelements (Na, K, Ca, Mg, P) and microelements (Fe, Mn, Cr, Se, Al and Zn) below tolerable upper intake level (ULs) for adults. The results of this manuscript found that there are no values for various chemical elements established by Codex Alimentarius for crude oils. The cold press extraction of babassu oil from a domestic machine yielded a high-quality oil, with no change in chemical composition, able to keep its natural antioxidants and stable to oxidation, resulting in a product that could be rapidly used by food and assist in the development of new studies on cold-pressed edible oils.

It was replaced by:

The edible oil extraction process has a growing interest because the final nutritional quality of the extracted oil depends on the procedure of its obtaining. In this context, a domestic cold oil press machine is a valuable tool that avoids chemical usage during oil extraction, in an environmentally friendly way. Although babassu (Attalea speciosa) oil is economically important in several Brazilian regions due to its nutritional and healthy features, few studies have been conducted on the chemical composition and stability of babassu oils extracted by cold pressing. Babassu oil major constituents are Saturated Fatty acids (~86.42 %), with the most prevalent fatty acids being Lauric (~47.40%), myristic (15.64%), and oleic (~11.28%) acids respectively, within the recommended range by Codex Alimentarius, presenting atherogenicity and thrombogenicity indexes favorable for human consumption. Peroxide value, Rancimat, and TGA/DSC results indicated that babassu oil is stable to oxidation. Also, macro- (Na, K, Ca, Mg, P) and micro-elements (Fe, Mn, Cr, Se, Al, and Zn) of babassu oil were determined, revealing levels below the tolerable upper intake level (ULs) for adults. These findings demonstrated that cold-press extraction using a domestic machine yielded a high-quality oil that kept oil chemical composition stable to oxidation with natural antioxidants.

_________________________________________________________________

Page 2, line 49

Is written:

The daily intake of oils in adequate amount plays an important role in keeping the body healthy [1,2].

It was replaced by:

Daily intake of vegetable oils in adequate amounts may play an essential role in keeping the body healthy [1,2].

_____________________________________________________________

Page 2, line 51

Is written:

…..considerably in several countries in recent years [3].

It was replaced by:

……….considerably worldwide in recent years [3].

_______________________________________________

Page 2, line 53

Is written

…… In order to better meet the health …..

It was replaced by:

……..In order to attend the health….

__________________________________________________________

Page 2, line 55

Is written:

….rapeseed mustered, cotton seed etc. [5],

It was replaced by:

………..rapeseed mustard, cottonseed and so on [5],

________________________________________________________________________________

Page 2, line 57

Is written:  …..yield

It was replaced by: ……yields

_____________________________________________________________________________________

Reviewer 1: I don't think that oxidative stability is a "process"... in my opinion, is a chemical property.

Page 2, line 60-61

Is written:

An important indicator of the quality of edible oils is its oxidative stability, which contributes to the evaluation of their shelf life [8-10]. Oxidative stability is a process influenced by the chemical composition of the oil and action of various factors with pro and antioxidant characteristics, it can occur during oil processing and storage [11].

It was replaced by:

A critical indicator of the quality of edible oils is its oxidative stability, which is directly related to their shelf life [8-10]. Oxidative stability is influenced by the chemical composition of the oil and action of various factors with pro and antioxidant characteristics, and it can occur during oil processing and storage [11].

______________________________________________________________________________________________

Page 2 line 66

Is written:   … decrease in oil shelf life [11].

Is was replaced by: …..decrease in the oil shelf life [11].

_______________________________________________________________________________________

Page 2 lines 70-74

Is written:  

 Brazilian Cerrado regions have a huge amount of plant species, in which they are conducive to

the growth and development of palm trees such as babassu, which can provide great nutritional

resources and financial return for the poorer population [15]. Scientifically known as Attalea speciosa

mart. Ex spreng, sinonym Orbignya paleratha and O. oleifera [16], the babassu nut is the second bestselling product in Brazil [16].

Is was replaced by:

Brazilian Cerrado regions have a huge variety of plant species and are conducive to the growth and development of palm trees such as babassu, which can provide great nutritional resources and financial return for the poorer population [15]. Babassu also grows in the western portion of South America. Babassu is scientifically known as Attalea speciosa mart. Ex spreng, sinonym Orbignya paleratha, and O. oleifera [16], and its nuts are the second bestselling product in Brazil [16].

_____________________________________________________________________________________________

Page 2, lines 77, 78, 79-82

Is written:

Several parts of babassu are used in the food human and animal, handicrafts, house coverings, cosmetics and fuel. Babassu oil extracted from nuts (endocarp) is used for human consumption [15].

The production of babassu in Brazil is of great importance for agroextractivism and is mainly  related to the large number of products and by-products derived from all parts of the plant (mesocarp, endocarp, epicarp and leaves). Several parts of babassu are used in the food human and

 animal, handicrafts, house coverings, cosmetics and fuel. Babassu oil extracted from nuts (endocarp)

is used for human consumption [15].

It was replaced by:

The production of babassu in Brazil is of great importance for agroextractivism and is mainly related to a large number of products and by-products derived from all parts of the plant (mesocarp, endocarp, epicarp, and leaves). Parts of babassu such as epicarp and endocarp are used in handicrafts, home coverings, burning in domestic and commercial ovens and organic manure. The uses for babassu oil extracted from nuts (endocarp) include human feed, cosmetics and fuel [15]. In addition, studies have shown the important activity of babassu oil microemulsion as a natural product to improve immune system function [17].

________________________________________________________________

Page 2, lines 80-86

Is written:

Over the recent time, studies have shown the important activity of antimicrobial potential of

 babassu palm leaf extract [17] and babassu oil microemulsion as a natural product to improve

 immune system function. According to research based in solvent extraction, the physicochemical

properties and fatty acid composition of babassu and indaiá (A. dubia) are similar in their

 composition of lauric acid, myristic acid, caprylic acid and capric acid [18]. However, the presence of metal in babassu nuts influenced

It was replaced by:

A Recent study demonstrated that the physicochemical properties and fatty acid composition of babassu and indaiá (A. dubia) are similar in their composition of lauric acid, myristic acid, caprylic acid, and capric acid when extracted by solvent extraction method [18]. However, studies on the mineral composition of various edible oils are scarce, especially the presence of macro and micro-elements in babassu oil, which may be influenced by agronomic, climatic and industrial applications [19].

_________________________________________________________________

Comment reviewer 1: The authors must indicate that a comparison of obtained values of fatty acids concentrations for Babassu oil and requiered (?) values for this oils by the Codex Alimentarius, are presented in table 1. In the text, a good explanation on the information/data contained in tables is crucial in a well-prepared manuscript.

Page 2, line 87-93

Is written:  

To date, as far as we know, only the composition of oils and fatty acids in cold pressed babassu  oil has been studied in the Brazilian Amazon [20], and there is no available data on macro- and  micro-elements composition in babassu oil. However, according to results obtained by Naozuka et al. [19], there are metals, metalloids and non-metals in the babassu nut and mesocarp. Thus, we  believe that babassu oil may also be contaminated with metals during the production process due to the presence of metal in nuts. Given the above, there are few studies on the properties and content of babassu oil in several regions of Brazil, especially in the Midwest region of the country.

It was replaced by:

To date, despite the physicochemical characterization of babassu oils has been performed, especially for determining the fatty acid composition of babassu oil obtained from Brazilian Amazon region [1], to the best of our knowledge, there is no available data on macro- and micro-elements composition in babassu oil extracted by cold press method. However, it is important to point out that Naozuka et al. [19] have demonstrated that there are metals, metalloids, and non-metals in the babassu nut and mesocarp. Thus, we believe it seems to indicate that babassu oil may also be contaminated with metals during the production process due to the presence of metals in nuts. In addition, there are few studies on the properties and content of babassu oil extracted from other Brazilian regions, especially the obtained from the Brazilian Cerrado located in the Midwest region of the country.

__________________________________________________________________________________________

Pag 3, lines 96

Is written:  

Taking this opportunity, companies....

It was replaced by:

To exploit this opportunity, companies

______________________________________________________________________________________

Pag 3, line 101-111

Is written

Motivated by the manuscript published by Serra et al. [20], which emphasizes the need for new forms of oil extraction and filtration methods, we present in our manuscript an unpublished study that considers an extraction method performed by marketable domestic extraction machines. The aim of this study is to evaluate for the first time the physicochemical properties, oxidative stability, optical properties, thermal stability and mineral composition of edible babassu oil consumed and collected in the Midwest region of Brazil and extracted by cold pressing using a domestic extraction machine. New results on thermal analysis, molecular absorbance (UV/VIS) and fluorescence, as well as quantification of macroelements and microelements present in babassu oil produced and consumed in the Midwest region of Brazil are presented in this study. In addition, the results will be compared with those values established by Codex Alimentarius [22] for babassu oil, crude oil and specialized literature.

it was replaced to:

Motivated by the manuscript published by Serra et al. [20], which emphasizes the need for new ways of oil extraction and filtration methods, the present investigation aimed to evaluate for the first time the physicochemical properties, thermal and oxidative stability, optical properties and mineral composition of edible babassu oil collected and consumed in the Midwest region of Brazil and extracted by cold pressing using a domestic extraction machine. Original results on thermal analysis, molecular absorbance (UV-VIS), and fluorescence, as well as quantification of macro- and micro-elements in babassu oil from the Midwest region of Brazil are presented in this study. Also, we compared the results of the fatty acids, and physicochemical profile with the values established by Codex Alimentarius [22] for vegetable oils. Furthermore, the mineral content obtained from babassu oil was compared to Codex Alimentarius values for refined oils and tolerable higher intake levels for adults (31 to 50 years) [23]. 

______________________________________________________________________________________

Page 3, lines 113

Is written

 Fatty acid composition

It was replaced by:

Sample moisture and yield

Observation: We have added production yield information as suggested by Reviewer 2.

2.1. Sample moisture and yield

The determination of moisture in seeds of babassu revealed that it contains 4.153 ± 0.030% of water. Low amounts of water content usually yield a larger amount of oil extraction, which can be explained not only by the chemical composition but as well because the water may act as a lubricant, decreasing the pressure in the compression area of the press [2].

Oil yield was 5.6 %, which means 265 ml from the starter material of 5 kg. It is worth to note that the yield can be increased by repressing the oil mass, which we did not do, once the primary goal of this paper was to evaluate the quality of the oil.

____________________________________________________________

Page 3, lines 114

Is written:

2.1. Fatty acid composition

It was replaced to:

2.2 Fatty acid composition

_________________________________________________________________

Page 3,  lines 119-121

Is written: Lauric acid (C12:0) was found to be the predominant  Saturated Fatty Acid (SFA) comprising 47.40% of total fatty acid composition. In addition, observed contents of myristic, palmitic and capric acid were above the reported data by Ref. [20].

It was replaced to:

Lauric oils and their derivatives have many applications in both the food and chemical industries. In addition, observed contents of myristic, palmitic and capric acid obtained from cold-pressed babassu oil using a domestic extraction machine was higher than those values of babassu oil using the artisanal cold pressing method using hydraulic presses [20].

_______________________________________________________________

Page 3,  lines 123 - 136

Is written:

In table 1, all fatty acid composition of babassu oil obtained by cold pressing extraction machine were within the values established by Codex Alimentarius [22] for babassu oil.  ……………. Table 1 show that the babassu oil is composed majorly of SFA (86.42%), followed by monounsaturated fatty acids (MUFA, 11.43%) and the least was seen in the polyunsaturated fatty acids (PUFA, 2.15%). In the study by Serra et al. [20] in the northern region of Brazil, the composition of SFA, MUFA and PUFA was 89.5%, 9.0% and 1.0%. After comparison, babassu oil from the Midwest region of Brazil with those from the Amazon studied by Serra et al. [20], it was found that there is a percentage difference between saturated fatty acids (3.56%), monounsaturated fatty acids (21.25%) and difference between polyunsaturated fatty acids (46.51%). Therefore, the obtained values in both studies do not show a particular pattern due to fact that fatty acids are dependent on temperature, soil and climate [24]. In the northeast region of Brazil, the study by Santos et al. [25] shows babassu oils of the same territorial range have similar chemical composition, regardless of the biome. However, the fatty acid composition results obtained by Santos et al. [25] are different from our results and those obtained by Serra et al. [20].

It was replaced to:

Table 1 presents a comparison between the obtained values of fatty acid concentrations of babassu oil and the required values by the Codex Alimentarius, revealing that the cold pressing extraction machine can be used to extract high-quality oil as the extracted oil meets the requirements established by Codex Alimentarius [22] for babassu oil. However, for butyric, undecylic, tridecanoic, and elaidic acid values have not yet established by Codex Alimentarius [22]. Table 1 also shows that the babassu oil is majorly composed of SFA (86.42%), followed by monounsaturated fatty acids (MUFA, 11.43%) and the least was seen in the polyunsaturated fatty acids (PUFA, 2.15%). Serra et al. [20] have reported that babassu oil obtained from the northern region of Brazil presented 89.5, 9.0 and 1.0% SFA, MUFA and PUFA, respectively. After comparison, babassu oil from the Midwest region of Brazil with those from the Amazon studied by Serra et al. [20], we found a percentage difference between saturated fatty acids, monounsaturated fatty acids and polyunsaturated fatty acids of 3.56%, 21.25%, and 46.51%, respectively. Therefore, the obtained values in both studies do not show a pattern since fatty acids are dependent on temperature, soil, and climate [24]. In the northeast region of Brazil, the study by Santos et al. [25] shows babassu oils obtained from the same territorial range have similar chemical composition, regardless of the biome. However, the fatty acid composition results obtained by Santos et al. [25] are different from our results and those obtained by Serra et al. [20].

___________________________________________________________________________________________

Page 4, line 140

Is written:

2.2 Atherogenicity and thrombogenicity index

It was replaced to:

2.3 Atherogenicity and thrombogenicity index

_________________________________________________________________________________________

Comment of revisor 1: This part of the text can be improved.

Pag 4, lines 144-146

Is written:

In table 1, the optimal nutritional quality indices were as follows: 8.72 for the atherogenicity index (AI), and 3.63 for the thrombogenicity index (TI). The high nutritional value of babassu oil is  primarily related to its saturated fatty acids (SFA) profile [7]. It is worth to note suggesting that the

 consumption of these babassu oil species could be of benefit to human health. It should be  mentioned that atherogenic (24.04) and thrombogenic (10.90) index calculated by us based on  coconut oil [26] is higher than the values for babassu oil obtained in our study, although both oils

 have the similar fatty acid profile.

It was replaced by:

Table 1 presents the nutritional quality indexes of babassu oil, which determined values are 8.72 and 3.63 for the atherogenicity index (AI) and thrombogenicity index (TI). In addition, the atherogenic and thrombogenic indexes in babassu oil are lower than the values for coconut oil (AI = 24.04 and TI =10.90, respectively) [26]. However, AI and IT for cold-pressed oil using a domestic extraction machine are within the range obtained for Amazon nut oil blends (AI = 0.1-14.6 and TI = 0, 18-6.69) [7]. The high nutritional value of babassu oil is primarily related to its saturated fatty acids (SFA) profile [7], suggesting that the consumption of these oils could be benefiting to human health.

______________________________________________________

Page 4, line 154

Is written:

According to Houston [28], the intake of saturated fatty acids

It was replaced to:

According to Houston [28], the correlation between the intake

_____________________________________________________________________

Page 5 line  168

Is written: …….products creating less health hazards

It was replaced to:… products creating few health hazards

______________________________________________________________________

Page 5, line169

Is written: 2.3 Chemical physical analysis

It was replaced to: 2.4 Chemical physical analysis

_________________________________________________________

Page 5, line177

Is written: …………….as it is one

It was replaced to: ……..because it is one

___________________________________________________________________

Page 5 lines  191-195

Is written: The acidity index represents the amount of free fatty acids after triglyceride hydrolysis [36]. It is recognized as the most important physicochemical index used to evaluate oil quality [9]. The acidity index of babassu oil (3.47) is lower than the value stipulated by Codex Alimentarius [22] (4 mg 194 KOH/g) and higher then found by Serra et al. [20] (1.06 mg KOH/g). Again, these results corroborate the stability on babassu oil.

It was replaced by: The acidity index represents the amount of free fatty acids after triglyceride hydrolysis [36]. Actually, the acidity index the most important physicochemical index used to evaluate oil quality [9]. The acidity index of babassu oil (3.47 mg KOH/g) is lower than the value stipulated by Codex Alimentarius [22] (4 mg KOH/g) (Table 2) and higher than found by Serra et al. [20] (1.06 mg KOH/g). Again, these results evidence the high stability of babassu oil.

___________________________________________________________________________

Page 6,  line 202

Is written: Saponification index represents the average..

It was replaced to: The saponification index represents the average

_____________________________________________________________________________

Page 6, line 203

Is written: …….of short and medium chain acids [20].

It was replaced to: ………. short and medium-chain acids [20].

_____________________________________________________________________

Page 6, line 209

Is written: The unsaponifiable

It was replaced to: Unsaponifiable

___________________________________________________________________

Page 6, line 213

Is written:  …….. to delay the ……..

It was replaced to: ……to delay in the …….

_____________________________________________________________________________

Page 6, line 215

Is written: 2.4 Oxidative stability assessment by Rancimat method

It was replaced to: 2.5 Oxidative stability assessment by Rancimat method

_____________________________________________________________________

Page 6, line 221

Is written:  …babassu nut oil, which can be attributed among other factors to the high degree of saturation (86.42%) of the oil, especially lauric fatty acid (47.40%). To increase the oxidative stability of oils, it is recommended to reduce oxidation promoting factors such as exposure to light, oxygen, heat and increased antioxidant content [12]. In fact, the main limiting factor for a food's shelf life is lipid oxidation [11].

It was replaced to: …..babassu nut oil, which can be due to the high degree of saturation (86.42%) of the oil, among other factors, especially lauric fatty acid (47.40%). It is recommended to reduce oxidation, promoting factors such as exposure to light, oxygen and, heatto increase the oxidative stability of oils. , as well as to intensify the antioxidant content in the oil [12]. The main factor that limits the shelf life of food is lipid oxidation [11].

_____________________________________________________________________

Page 6, line 229

Is written: 2.4 Thermogravimetry / Derivative Thermogravimetry (TG/DTG)

It was replaced to: 2.6 Thermogravimetry / Derivative Thermogravimetry (TG/DTG)

____________________________________________________________________________________

Page 6, line 230-231

Is written: (AR)

It was replaced to: (Air)

 ______________________________________________________________________

Page 7, line 244

Is written: To allow clarity of the steps of decomposition,

It was replaced to: Aiming to clarify the steps of decomposition,

________________________________________________________________________________

Page 7, line 246

Is written: … TG and DTG are listed in Table 3

It was replaced to: … TG and DTG are presented in Table 3

_____________________________________________________________________

Page 7, line 254

Is written:…..three steps of mass loss described below: as can

It was replaced to: ……three steps of mass loss, as following described. As can

______________________________________________________________________

Page 7, line 255

Is written:  …..decomposition occurs at the Tonset of 255.68 °C, there was a mass.

It was replaced to: …decomposition occurs at the Tonset of 255.68 °C, in which there was a mass

______________________________________________________________________

page 7,  line 258

Is written: ….loss and endset of the 555.22 °C.

It was replaced to:  endset of the 555.22 °C occurred.

________________________________________________________________________

page 8,  line 259

Is written: losses in stages 2 and 3 are attributed

It was replaced to: losses in stages 2 and 3 happen when higher

_____________________________________________________________________________

Page 8, 267

Is written:  Figure 3. … oil in oxidative …

It was replaced to: Figure 3. ………. oil in an oxidative……

___________________________________________________________________________

Page 8, 273-274

Is written: antioxidants [43]. In fact, in the process of refining of oils there is a reduction in total content of phenol, β-carotene and oxygen radical absorbance [44].

It was replaced to: In fact, the process of refining of oils promotes the reduction in the total content of phenol, β-carotene, and oxygen radical absorbance [44].

____________________________________________________________

Page 8,  line 275

Is written: Table 3. Results obtained from the TG/DTG curve of babassu oil in a synthetic air and nitrogen atmosphere under dynamic and almost isothermal conditions

It was replaced to: Table 3. Results obtained from the TG/DTG curve of babassu oil in a synthetic air (Air) and nitrogen (N2) atmosphere under dynamic and almost isothermal conditions

Is written in table 3: ar

It was replaced to: Air

____________________________________________________________________________

Page 9,  line 279

Is written:…….. in atmosphere of nitrogen (N2) with flow rate of 50 mL min-1

It was replaced to: ….. in an atmosphere of nitrogen (N2) with a flow rate of 50 ml min-1

______________________________________________________________________________________________________________

Page 9, lines 282

Is written:

According to Zhang et al. [45] and Tan et al. [46], the crystallization profile in oils is  characterized by the beginning of fat crystal formation, which is related is greatly influenced by mass transfer, heat transfer, cooling rate, viscosity, presence of shear, etc [47]. In fact, the crystallization of fatty acids is a rearrangement of molecules due to the presence of saturated triglycerides. In their results, the author [45], shows the isothermal photomicrographs of crystals formed from Palm Oil on various temperatures

It was replaced to:

According to Zhang et al. [45] and Tan et al. [46], the crystallization profile in oils are characterized by the beginning of fat crystal formation, which is related and greatly influenced by mass transfer, heat transfer, cooling rate, viscosity, presence of shear, etc. [47]. Zhang et al. [45] demonstrated that the isothermal photomicrographs of crystals formed from Palm Oil on various temperatures. Actually, the crystallization of fatty acids is a rearrangement of molecules due to the presence of saturated triglycerides.

_________________________________________________________________________

Page line 289

Is written: Figure 4. Figure 4. Cooling of babassu oil on oxidative atmosphere of nitrogen under almost quasi-isothermal condition.

It was replaced to:  Figure 4. Cooling of babassu oil on an oxidative atmosphere of nitrogen under almost quasi-isothermal conditions.

________________________________________________________________________________

Page 9, line 292

Is written:  ……. from -30 to -35 °C at rate of..

It was replaced to: ……..from -30 to -35 °C at a rate of…

_______________________________________________________________________________________

Page 9, line 293

Is written: …..in the present study was bigger than

It was replaced to: present study was higher than the

________________________________________________________________

Page 10,

Is written: 2.5 Molecular Absorption Spectroscopy (UV/VIS) and Molecular 300 Fluorescence

It was replaced to: 2.7 Molecular Absorption (UV-VIS) and Molecular Fluorescence Spectroscopy

_____________________________________________________________________

Page 10, line 301

Is written:  Figure 5 shows the UV-Vis absorbance spectra of babassu oil..

It was replaced to: Figure 5 shows the UV-VIS absorbance spectrum of babassu oil..

__________________________________________________________________________

Page 10, line 304

Is written: .. In Fig.5.b, the second spectral ranges studied from 350 to 450 nm correspond to chlorophylls or carotenoids [50,51].

It was replaced to: ..Besides, the second spectral ranges observed between 350  and 450 nm, as presented in the inset of Figure 5  corresponds to chlorophylls or carotenoids [50, 51].

_____________________________________________________________________

Page 10, lines 308 -310

Is written:  Figure 5 a) UV-VIS absorption spectrum (wavelength from 210 to 500 nm) of babassu oil diluted in  hexane at different concentrations (0.0010; 0.0005; 0.0100; 0.0500 gL-1). b) Spectral magnification (wavelength from 361 to 470 nm).

It was replaced to: Figure 5.  UV-VIS absorption spectrum (wavelength from 210 to 500 nm) of babassu oil diluted in hexane at 0.0100 gL-1). Inset Zoom of the UV-Vis spectrum in the 350 to 500 nm range.

___________________________________________________________________

Page 10,  lines 311-312

Ias written:In the literature, the presence of bands in vegetable oils is used as a quality parameter and monitoring of oxidation products [52].

It was replaced to: Recent studies have demonstrated that the molecular absorption bands of vegetable oils, in the UV-Vis range, can be used as a quality parameter for monitoring the oil oxidation [52, 53]. The typical excitation-emission fluorescence map of babassu oils is shown in Figure 6.

_________________________________________________________________________________________

Page 10, lines 313-314

Is written:… spectra for diluted in 1 x 10-314 g/ml and 0.05 g/ml samples are shown.

It was replaced to : ….spectra of the oil diluted at 1 x 10-3 g/ml are presented.

_____________________________________________________________________________________

Page 10, lines: 314-318,

Is written: In the literature, the presence of bands in vegetable oils is used as a quality parameter and monitoring of oxidation products [52]. Typical fluorescence spectra of babassu oils are shown in Figure 6. The fluorescence depends on sample concentration; therefore, spectra for diluted in 1 x 10-3 g/ml and 0.05 g/ml samples are shown. Both total fluorescence and total synchronous spectra are presented for the same oils, to enable comparison. The total fluorescence spectrum of babassu oils measured exhibits two intense bands, one with excitation at about (275-300) nm and emission at about 300–325 nm and the second with excitation at about (300 e 325) nm and emission at about 425–525 nm (450-500), Figure 6. The fluorescence bands may be associated with antioxidants present

Page 11, lines 319-310 in babassu vegetable oil, ie the presence of alpha-tocopherol and/or the presence of  carotenoid group antioxidants [51].

It was replaced to:

Page 10, lines 319-320

Recent studies have demonstrated that the molecular absorption bands of vegetable oils, in the UV-Vis range, can be used as a quality parameter for monitoring the oil oxidation [52, 53]. The typical excitation-emission fluorescence map of babassu oils is shown in Figure 6. The fluorescence depends on sample concentration; therefore, spectra of the oil diluted at 1 x 10-3 g/ml and 0.05 g/ml are presented. The total fluorescence spectrum of babassu oils (Figure 6) exhibits two intense bands, one with excitation at about 275–300 nm and emission at about 300–325 nm and the second with excitation at about 300–325 nm and emission at about 425–525 nm, Figure 6. These fluorescence bands may be associated with endogenous antioxidants present in babassu vegetable oil, i.e., the presence of tocopherols and/or carotenoids [51].

_____________________________________________________________________________________

Page 11, line 323

Is written: … correspond to chlorophylls [50].

It was replaced to: corresponds to carotenoids [50].

_________________________________________________________________________

Page 11, lines 324-327

is written: On the other hand, the spectral ranges studied λ = 425-525 nm have been attributed to range of fatty acid oxidation products and tocopherols. In fact, the fluorescence spectra in the emission from 400 to 500 nm have been attributed to fatty acid oxidation products and tocopherols presents in blended oils [54].

It was replaced to: However, the emission observed in the λ = 425–525 nm range can also be correlated to the range of fatty acid oxidation products [49]. The fatty acid oxidation products and tocopherol present in blended oils is responsible for the fluorescence spectra in the emission from 400 to 500 nm [55].

_________________________________________________________________________

Page 11, line 335

Is written: 2.6 Determination of macro- and micro-elements

It was replaced to: 2.8 Determination of macro- and micro-elements

_____________________________________________________________________

Page 11, line 342

Is written:  (R2) for each element studied in this manuscript.

It was replaced to: (R2) for each element quantified in the present study

________________________________________________________________________________________

Page 11, lines 344-346

Is written: ….. oil extraction and treatment process [1, 55].

It was replaced to:  as well as exogenous sources that contaminated the oil during technique of processing such as extracting, crushing, refining, bleaching, hydrogenation and deodorisation [1, 56].

__________________________________________________________________________________________

Pag.11, line 347-353

Is written:    

The quality of edible oils has been monitored based on the presence of elements such as Ag, As, Ba, Be, Cd, Co, Cr, Cu Fe, Hg, Mn, Mo, Ni, Pb, Sb, Ti, Tl and V in Spain [1]; Pb, Cd, Ni, Mn, Zn, Cu, Fe, Ca and Mg in Iran [56], and Cu, Zn, Fe, Mn, Cd, Ni, Pb, and As in China [13], Fe, Mn, Zn, Cu, Pb, Co, Cd, Na, K, Ca and Mg in Turkey [57] and Pb, Cd, Ni and As in olive oil produced in Italy [58]. In the above results, values of contents obtained in the Ref. [13, 56-58] were compared with those studies established by requirements for chemical elements. Thus, theses studies have shown that macro- and micro-elements can be beneficial or may be harmful to human health.

It was replaced to:

It is well known that analysis of the elemental composition of vegetable oils can be used to monitor their quality, adulteration, and conservation of their products. For example, the determination of Ag, As, Ba, Be, Cd, Co, Cr, Cu Fe, Hg, Mn, Mo, Ni, Pb, Sb, Ti, Tl, and V has been used to evaluate the quality of virgin olive, olive, pomace-olive, sunflower, soybean and corn oils in Spain [1]; In Iran, the content of Pb, Cd, Ni, Mn, Zn, Cu, Fe, Ca and Mg were investigated to analyze olive, canola, sunflower and soybean oils [57]. Levels of Cu, Zn, Fe, Mn, Cd, Ni, Pb, and As were also used for monitoring edible oils, such as soybean, corn, peanut, sesame, rapeseed, cottonseed, olive, blend and sunflower oils in China [13]. In Turkey, olive, hazelnut, sunflower oils as well as margarine and butter were tested by means of the presence of Fe, Mn, Zn, Cu, Pb, Co, Cd, Na, K, Ca and Mg [58] while Pb, Cd, Ni and As has been quantified for quality control of olive oils produced in Italy [59]. In summary, these investigations have demonstrated that the potential health risks and/or health benefits for human consumption can correlate with the presence of macro- and microelement in edible oils. The elemental content in edible oils is useful in determining the quality of refined oils when comparing the values ​​set by Codex Alimentarius [22] and also compared to tolerable upper intake levels for adults [23].

________________________________________________________________________________

Page 12, line 362

Is written:  In Table 4, the iron concentration..

It was replaced to:  Table 4 presents that the iron concentration

_______________________________________________________________________________________

Page 12, line 363

Is written: Iron and copper present in the oil react directly …

It was replaced to: Iron and copper present in the oil may react directly…

_______________________________________________________________________________________

Page 12, lines 368-369

Is written: In addition, trace elements reduce product shelf life and alter the nutritional value of the oil [56].

It was replaeced to: Also, trace elements reduce product shelf life and alter the nutritional value of the oils [56].

____________________________________________________________________________________

Page 12, line 374

Is written:  Results obtained within the framework of this study (Table 4).

It was replaced to: We compared the results obtained within the framework of this study (Table 4) to…

Page 12, line  381

Comment revisor 1: This sentence is wrong... nobody has ingested babassu oil for this study. Babassu oil contains macro- and micro-nutrients with levels below tolerable values for an oil for human feed. The authors must pay special attention to the redaction, otherwise they will confuse the readers.

Page 12, line  381

Is written:

.., it is found that ingestion of the babassu oil are below the values tolerable upper intake level, thus, do not pose a health hazard risk regarding elemental concentration [23], being safe for human consumption

It was replaced by:

…, it is found these content in babassu oil are below the values tolerable upper intake level. Consequently, the present study demonstrated that the babassu oil does not pose a health hazard risk regarding elemental concentration [23], being safe for human consumption.

___________________________________________________________________________________________

Pag.13, line 391

Is written:

….. (R2), and analytical

It was replaced by: ….. (R2), and analytical

___________________________________________________________________________________________

Page 14 line 406

Is written:  Several ripe babassu fruits…...

It was replaced to: We collected several ripe babassu fruits……

____________________________________________________________________________________________

Page 14, line 407

Is written:  The babassu fruit

It was replaced to: The harvest of babassu fruit

____________________________________________________________________________________________

Page 14, line 412

Is written:   ……..extraction.

It was replacet to:  extraction and water content.

___________________________________________________________________________________________

Page 15, line 430

Is written: … The following ..

It was replaced to: ... We used the…

__________________________________________________________________________________________

Page 16, lines 453-454

Is written: , unsaponifiable matter (Ca 6a-40) and refractive index (Cc 7-25). Assays were performed in triplicate

It was replaced to:  , unsaponifiable matter (Ca 6a-40) and refractive index (Cc 7-25).

___________________________________________________________________________________________

Page 16, line 477

Is written:….. In this paper the thermal stability was evaluated by the Onset Point Temperature (Tonset). The Onset Point is obtained by the extrapolated beginning of the curve, being defined by the point of intersection of the tangent with the point of maximum slope, on the principal side of the mass loss curve with the base line extrapolated (TA Advantage/Universal Analysis Software).

It was replaced to: … In this paper, we determined the thermal stability by the Onset Point Temperature (Tonset). The obtaining of the Onset Point is by the extrapolated beginning of the curve, which is defined by the point of intersection of the tangent with the point of maximum slope, on the principal site of the mass loss curve with the baseline extrapolated (TA Advantage/Universal Analysis Software).

____________________________________________________________________________________________

Page 16, line 481

Is written:

3.7 Molecular spectroscopy visible (UV/VIS) and fluorescence

It was replaced by:

3.7 Molecular spectroscopy visible (UV-VIS) and fluorescence

_____________________________________________________________________________________________

Page 16, line 483

Is written: .. 265UV/Vis,

It was replaced by: .. 265UV/VIS,

_____________________________________________________________________________________________

Page 16, lines 482-483

Is written:  Samples of babassu oils were diluted separately in HPLC grade hexane at a concentration of 10 483 g/L.

It was replaced to: Babassu oil was diluted in HPLC grade hexane at a concentration of 10 g/L and from stock solution, we prepared different dilutions for spectra reading at 1 x 10-3 g/ml and 0.05 g/ml.

_____________________________________________________________________________________________

Page 16,  line 485

Is written: . The UV-visible absorption spectra were………… All analyzes were performed

It was replaced to:  The collection of UV-Vis absorption spectra as………….. All analyses were..

______________________________________________________________________________________________

Page 17, line 494

Is writeen: ……were weighted

It was replaced to: ……was weighted

______________________________________________________________________________________________

Page 17, line 496

Is written:  .A microwave system

It was replaced to: ..The oil samples digesting occurred in a microwave

______________________________________________________________________________________________

Page 17, line 508

Is written:  For the quantitative analysis of oils calibration curves were build on five different concentrations.

It was replaced to: We used five different concentrations to build calibration curves for the

___________________________________________________________________________________________

In the page 17 was added the sentences:

The values of fatty acid composition, acidity index, peroxide index, saponification index of babassu oil extracted by cold pressing using a domestic extraction machine were above the artisanal methods by cold pressing using hydraulic presses.

___________________________________________________________________________________________

Page 18, line 525

Is written: …., once the formation of hazardous products is minimized by its stability.

It was replaced to:  once its stability minimizes the formation of hazardous products.

Reviewer 2 Report

The presented studies concern oxidative stability and elemental analysis of babassu edible oil produced in domestic extraction machine. The paper is interesting but it concerns is local issue.  In addition, the scientific purpose is unclear in light of the conclusions. The conclusions state that babassu oil produced in domestic extraction machine is healthier than those oils extracted by solven or hot press techniques. This conclusion is formulated without comparison with babassu oil produced by other methods. The aim of the work or conclusion should be rewritten.

Minor remarks,

oil production yield should be added,

in figures 2 and 3, the solid line designation with the Ar abbreviation suggests that this curve was measured in an argon atmosphere - please correct it ,

the basic parameters of harvested fruits such as water content etc. should be added.

Author Response

Reviewer 2

All grammar suggestions over the text have been accepted, and we agree that the text needed improvement, so we thoroughly checked the text. Thank you very much for your valuable suggestion. We have replaced it with new figure with better understanding. Hopefully you will find it justified.

The manuscript has been revised as per the comments given by the reviewer, and our responses to all the comments are as follows:

Reviewer 2:   The paper is interesting but it concerns is local issue.

Answers to the reviewer:  Dear reviewer, this palm tree is found in various parts of Latin America.

We added a sentence to emphasize that this palm tree exists in all of Latin America (See below).

Page 2 lines 70-74

Is written:  

 Brazilian Cerrado regions have a huge amount of plant species, in which they are conducive to the growth and development of palm trees such as babassu, which can provide great nutritional resources and financial return for the poorer population [15]. Scientifically known as Attalea speciosa mart. Ex spreng, sinonym Orbignya paleratha and O. oleifera [16], the babassu nut is the second bestselling product in Brazil [16].

Is was replaced by:

Brazilian Cerrado regions have a huge variety of plant species and are conducive to the growth and development of palm trees such as babassu, which can provide great nutritional resources and financial return for the poorer population [15]. Babassu also grows in the western portion of South America. Babassu is scientifically known as Attalea speciosa mart. Ex spreng, sinonym Orbignya paleratha, and O. oleifera [16], and its nuts are the second bestselling product in Brazil [16].

Questions reviewer 2:   In addition, the scientific purpose is unclear in light of the conclusions.

Answers to the reviewer:  Thank you for your comment. Thus, we add more information to the conclusions.

Is written:

Babassu oils have important fatty acids, embracing several groups of SFAs, 519 MUFAs, and at  minor level, PUFAs. Lauric and myristic acid were found to be the predominant SFAs in composition and therefore play a major role in contributing for babassu oil stability. According results, index of atherogenicity, and index of thrombogenicity obtained from babassu oils are favorable for consumption. Results of the levels of peroxide and Rancimat method found in babassu oil indicate that this oil may be stable to oxidation, which favors its use to cooking or using as a frying oil, once the formation of hazardous products is minimized by its stability. These results were

supported by TGA/DSC outcomes on stability of the babassu oil in different atmosphere and temperatures, besides providing information on the thermal events such as melting and crystallization determining accurate and precise transition temperatures. Differences in the content of total fatty acids, of saturated and of mono-unsaturated fatty acids of babassu oil can be explained by the type of soil and climate of where the seed was extracted. The cold press method was able to keep important features from the crude oil, such as the presence of natural antioxidants (tocopherols, phenolic compounds and chlorophylls), contributing for the final product quality.

For the first time, macro- and micro-elements in the cold-pressed babassu oil were quantified. Results showed that the babassu oil contained levels of macro- (Na, K, Ca, Mg, P) and micro-elements (Fe, Mn, Cr, Se, Al and Zn) below tolerable upper intake level (ULs) for adults. In fact, obtaining cold pressed oils using a home extraction machine can provide a low concentration of  contaminating metals in the oils avoiding contamination due to refinement processes and use of chemical agents.

The results of this manuscript found that there are no values for some chemical elements (Na, K, Mg, Mn, Ni, Co, Cr, Se, Al, Cd, Mo and Zn) established by Codex Alimentarius Codex for crude oils.

From these results, it could be concluded that the analyzed babassu oil could be used as oils for daily consumption. However, the values of the composition of babassu oil fatty acids from Midwest region differ those from babassu oil in the Amazon region. Babassu oil fatty acid composition values are within international recommendations established by Codex Alimentarius.

The cold pressed extraction of babassu oil in a domestic extraction machine presented a high quality edible oil, stable to oxidation, with below concentrations of metals, yielding a final product able to keep its natural antioxidants and as result, a healthier than those oils extracted by solvent or hot press techniques, at a lower cost. The high-quality oil from Midwest babassu is a valuable resource and has potential for industrial use, such as Amazonian oils.

The data collected are important for monitoring purposes and, also to fill the gaps about quality control edible oils. This data may be useful for the food industries and assist in the development of new studies on cold-pressed edible oils.

It was replaced to:

Babassu oil is composed of saturated and unsaturated fatty acids, embracing several groups of SFAs, MUFAs, and at a minor level, PUFAs. Lauric and myristic acid was found to be the predominant SFAs, playing a major role in contributing to babassu oil stability. The values of fatty acid composition, acidity index, peroxide index, saponification index of babassu oil extracted by cold pressing using a domestic extraction machine were above the artisanal methods by cold pressing using hydraulic presses. According to the atherogenicity and thrombogenicity indexes, the obtained results demonstrated that babassu oil is favorable for consumption. Results of the levels of peroxide and Rancimat method revealed that babassu oil is very stable to oxidation, which favors its use to cooking or using as a frying oil, once its stability minimizes the formation of hazardous products. These assumptions are also supported by the findings from TGA/DSC data on the stability of the babassu oil in different atmosphere and temperatures, besides providing information on the thermal events such as melting and crystallization determining accurate and precise transition temperatures. Differences in the content of total fatty acids, of saturated and of mono-unsaturated fatty acids of babassu oil can be explained by the type of soil and climate of where the seed was collected. The cold press method preserved the important features of crude oil, such as the presence of natural antioxidants (tocopherols, phenolic compounds, and carotenoids), contributing to the final product quality.

In addition, for the first time, macro- and micro-elements of babassu oil extracted by cold-pressed were quantified. The results showed that babassu oil contained levels of macro- (Na, K, Ca, Mg, P) and micro-elements (Fe, Mn, Cr, Se, Al, and Zn) below the tolerable upper intake levels (ULs) for adults. In fact, obtaining cold-pressed oils using a home extraction machine can provide a low concentration of contaminating metals in oils, avoiding contamination due to refinement processes and the use of chemical agents.

From these results, we may conclude that the obtained babassu oil can be used as oil for daily consumption. However, the values of the composition of babassu oil fatty acids from the Midwest region differ from those from the Amazon region. Babassu oil fatty acid composition ​​is within international recommendations established by Codex Alimentarius.

The cold-pressed extraction of babassu oil using a domestic extraction machine presented a low-cost way to obtain a high-quality edible oil, stable to oxidation, with low concentrations of metals, yielding a final product able to keep its natural antioxidants and, consequently, a healthier oil than those oils extracted by solvents or hot press techniques.

Furthermore, the present investigation also demonstrated that babassu oil contains different macro- and micro-elements, such as Na, K, Mg, Mn, Ni, Co, Cr, Se, Al, Cd, Mo, and Zn, which is not yet established by Codex Alimentarius for crude oils.

Finally, it is worth to point out that the present findings improve the knowledge regarding the physicochemical characterization and nutritional content of babassu oil as well as reveal a few gaps to be filled in the quality control of edible oils. Additionally, the presented data may be useful for the food industry and assist in the development of new studies on cold-pressed edible oils.

 Reviewer 2:   The conclusions state that babassu oil produced in domestic extraction machine is healthier than those oils extracted by solvent or hot press techniques.

Answers to the reviewer:  Dear reviewer, we agree with your observation that the abstract is missing information. However we follow the rules of the "Molecules Journal". According to Molecules, MPDI, the abstract should be a total of about 200 words maximum.

However as suggested by reviewer 1, we made some changes in the abstract.

Is written:

Abstract: People are becoming more interested in healthier edible oils and natural foods. This is where the domestic oil press machine is becoming a valuable tool. Babassu oils are important for human nutrition and economics in several Brazilian regions. Nevertheless, few studies have been

conducted on the composition of babassu oils extracted by cold pressing. The aim of this manuscript was to study the physicochemical properties, oxidative stability, optical properties, thermal stability and mineral composition of babassu edible oil consumed in the Midwest region of Brazil and extracted by cold pressing using a domestic extraction machine. Babassu oils is comprised mainly by Saturated Fatty acids, (86.42 %), followed by monounsaturated fatty acids (MUFAs), (11.43%) and Polyunsaturated fatty acids (PUFAs) (2.15%). Lauric (47.40%), myristic (15.64%), oleic (11.28%) acids are the major fatty acids in the oil composition, with values within form those proposed international recommendations established by Codex Alimentarius. Index of atherogenicity and index of thrombogenicity shows babassu oils are favorable for consumption. Results of the levels of peroxide, Rancimat method and TGA/DSC in babassu oil indicate that this oil is stable to oxidation. According to absorbance and fluorecence results, babassu oils contain fluorophores as tocopherols, phenolic compounds, and chlorophylls. Babassu oil contained levels of macroelements (Na, K, Ca, Mg, P) and microelements (Fe, Mn, Cr, Se, Al and Zn) below tolerable upper intake level (ULs) for adults. The results of this manuscript found that there are no values for various chemical elements established by Codex Alimentarius for crude oils. The cold press extraction of babassu oil from a domestic machine yielded a high-quality oil, with no change in chemical composition, able to keep its natural antioxidants and stable to oxidation, resulting in a product that could be rapidly used by food and assist in the development of new studies on cold-pressed edible oils.

It was replaced to:

Abstract: The edible oil extraction process has a growing interest because the final nutritional quality of the extracted oil depends on the procedure of its obtaining. In this context, a domestic cold oil press machine is a valuable tool that avoids chemical usage during oil extraction, in an environmentally friendly way. Although babassu (Attalea speciosa) oil is economically important in several Brazilian regions due to its nutritional and healthy features, few studies have been conducted on the chemical composition and stability of babassu oils extracted by cold pressing. Babassu oil major constituents are Saturated Fatty acids (~86.42 %), with the most prevalent fatty acids being Lauric (~47.40%), myristic (15.64%), and oleic (~11.28%) acids respectively, within the recommended range by Codex Alimentarius, presenting atherogenicity and thrombogenicity indexes favorable for human consumption. Peroxide value, Rancimat, and TGA/DSC results indicated that babassu oil is stable to oxidation. Also, macro- (Na, K, Ca, Mg, P) and micro-elements (Fe, Mn, Cr, Se, Al, and Zn) of babassu oil were determined, revealing levels below the tolerable upper intake level (ULs) for adults. These findings demonstrated that cold-press extraction using a domestic machine yielded a high-quality oil that kept oil chemical composition stable to oxidation with natural antioxidants.

Reviewer 2:   This conclusion is formulated without comparison with babassu oil produced by other methods. The aim of the work or conclusion should be rewritten.

Answers to the reviewer:  Thank you for your comment. Thus, we add more information to the conclusions.

Reviewer 2:    Minor remarks,

Answers to the reviewer:  Answers to the reviewer:  all your suggestions were accepted by authors.

Reviewer 2:   oil production yield should be added,

Answers to the reviewer: we add information about production yield

Page 3, lines 113

2.1. Sample moisture and yield

The determination of moisture in seeds of babassu revealed that it contains 4.153 ± 0.030% of water. Low amounts of water content usually yield a larger amount of oil extraction, which can be explained not only by the chemical composition but as well because the water may act as a lubricant, decreasing the pressure in the compression area of the press [2].

Oil yield was 5.6 %, which means 265 ml from the starter material of 5 kg. It is worth to note that the yield can be increased by repressing the oil mass, which we did not do, once the primary goal of this paper was to evaluate the quality of the oil.

Page 14, 415

Answers to the reviewer: We added the following sentence in the methodology: For moisture determination, we homogenized and weighted 5 g of babassu seeds, and kept in oven at 105 o C, being hourly weighted after rest in desiccator until achievement of constant weight.

Reviewer 2:   in figures 2 and 3, the solid line designation with the Ar abbreviation suggests that this curve was measured in an argon atmosphere - please correct it ,

Answers to the reviewer: in figures 2 and 3, as well as text we change Ar to Air.

Page 6, line 230-231

Is written: (AR)

It was replaced to: (Air)

Reviewer 2:   the basic parameters of harvested fruits such as water content etc. should be added.

Answers to the reviewer: We added the following sentence in the methodology: For moisture determination, we homogenized and weighted 5 g of babassu seeds, and kept in oven at 105 o C, being hourly weighted after rest in desiccator until achievement of constant weight.

Round 2

Reviewer 1 Report

The manuscript “First study on oxidative stability and elemental analysis of babassu (Attalea speciosa) edible oil produced in Brazil using a domestic extraction machine”, by Elaine Melo, Flavio Michels, Daniela Arakaki, Nayara Lima, Daniel Gonçalves, Leandro Cavalheiro, Lincoln Oliveira, Anderson Caires, Priscila Hiane and Valter Nascimento, presents interesting information on the composition of Brazilian babassu oil extracted by cold pressing. Characteristics like oxidative stability and thermal stability were also measured. This study was carried out in order to determine whether or not babassu oil can be used in human feed, therefore, quality parameters of Codex Alimentarius were considered for comparisons.

The quality of the manuscript was improved respect the latest version. Acclaratory notes were incorporated by the authors (as suggested) and this contributed to a better understanding of the contents of the manuscript. I conclude that this paper can be accepted in its current form for publication in Molecules.